# A complex genetic interaction implicates that phospholipid asymmetry and phosphate homeostasis regulate Golgi functions

**Mamoru Miyasaka[1,2], Tetsuo Mioka[1], Takuma Kishimoto[1], Eriko Itoh[1], Kazuma Tanaka[1]***

**1** Division of Molecular Interaction, Institute for Genetic Medicine, Hokkaido University Graduate School of Life Science, Sapporo, Hokkaido, Japan, **2** Department of Gastroenterological Surgery II, Hokkaido University Faculty of Medicine, Sapporo, Hokkaido, Japan

* k-tanaka@igm.hokudai.ac.jp

**Data Availability Statement:** All relevant data are within the paper and its Supporting Information files.

## Abstract

In eukaryotic cells, phospholipid flippases translocate phospholipids from the exoplasmic to the cytoplasmic leaflet of the lipid bilayer. Budding yeast contains five flippases, of which Cdc50p-Drs2p and Neo1p are primarily involved in membrane trafficking in endosomes and Golgi membranes. The *ANY1/CFS1* gene was identified as a suppressor of growth defects in the *neo1Δ* and *cdc50Δ* mutants. Cfs1p is a membrane protein of the PQ-loop family and is localized to endosomal/Golgi membranes, but its relationship to phospholipid asymmetry remains unknown. The *neo1Δ cfs1Δ* mutant appears to function normally in membrane trafficking but may function abnormally in the regulation of phospholipid asymmetry. To identify a gene that is functionally relevant to *NEO1* and *CFS1*, we isolated a mutation that is synthetically lethal with *neo1Δ cfs1Δ* and identified *ERD1*. Erd1p is a Golgi membrane protein that is involved in the transport of phosphate (Pi) from the Golgi lumen to the cytoplasm. The Neo1p-depleted *cfs1Δ erd1Δ* mutant accumulated plasma membrane proteins in the Golgi, perhaps due to a lack of phosphatidylinositol 4-phosphate. The Neo1p-depleted *cfs1Δ erd1Δ* mutant also exhibited abnormal structure of the endoplasmic reticulum (ER) and induced an unfolded protein response, likely due to defects in the retrieval pathway from the *cis*-Golgi region to the ER. Genetic analyses suggest that accumulation of Pi in the Golgi lumen is responsible for defects in Golgi functions in the Neo1p-depleted *cfs1Δ erd1Δ* mutant. Thus, the luminal ionic environment is functionally relevant to phospholipid asymmetry. Our results suggest that flippase-mediated phospholipid redistribution and luminal Pi concentration coordinately regulate Golgi membrane functions.

## Introduction

The lipid composition of the plasma membrane (PM) in eukaryotic cells differs between the inner (cytoplasmic) leaflet and outer (exoplasmic) leaflet. Generally, phosphatidylcholine (PC) and sphingolipids distribute in the outer leaflet, whereas phosphatidylserine (PS) and phosphatidylethanolamine (PE) distribute in the inner leaflet [1]. This asymmetrical

**Funding:** This work was supported by JSPS KAKENHI Grant numbers JP18K14645 (TM), JP18K06104 (TK), and JP19K06536 (KT). The funders had no role in study design, data collection and analysis, decision to publish, or preparation of the manuscript.

**Competing interests:** The authors have declared that no competing interests exist.

distribution of phospholipids across the bilayer is conserved in yeast as well as higher forms of life. Changes in phospholipid asymmetry involve various cellular functions. For instance, exposure of PS to the outer leaflet of the PM signals the onset of blood coagulation and apoptosis and triggers phagocytosis [2], [3]. Cellular polarity is also induced due to local transbilayer changes in PE and PS levels in the PM [4], [5], [6].

P4-ATPases, termed flippases, are essential to maintain an asymmetric PM structure. These membrane proteins control the distribution of bilayer lipids by flipping phospholipids across the lipid bilayer from the outer to the inner leaflet [7]. Select flippases are localized to endosomes and Golgi membranes to regulate membrane trafficking that occurs between the membranes [8], [9]. In the budding yeast *Saccharomyces cerevisiae*, five flippases can be found: Drs2p, Dnf1p, Dnf2p, Dnf3p, and Neo1p [10]. These flippases, except for Neo1p, form complexes with a member of the CDC50 protein family [11]. Drs2p, Dnf1p/Dnf2p, and Dnf3p form complexes with Cdc50p, Lem3p and Crf1p, respectively. Further, these five flippases functionally overlap, but *NEO1* is an essential gene on its own [12]. The Cdc50p-Drs2p complex is localized to endosomes and the *trans*-Golgi network (TGN), and its functions in membrane trafficking have been extensively characterized. It has been proposed that phospholipid flipping by the Drs2p-Cdc50p complex promotes the formation of transport vesicles, including the clathrin-coated vesicle, but its mechanism remains unclarified [8], [9], [13]. Neo1p is also localized at endosomes and Golgi membranes to regulate membrane trafficking. Neo1p is required for retrograde transport from the *cis*-Golgi to the endoplasmic reticulum (ER) [14]. Neo1p is also involved in endocytosis and the vacuolar sorting pathway [15]. Flippase activity of Neo1p has not yet been demonstrated, but the *neo1* mutant expresses loss of PE asymmetry in the PM [16]. In addition, *NEO1* was isolated as a multicopy suppressor of the *cdc50Δ* mutation [11], and Neo1p has similar functions to the Cdc50p-Drs2p complex in terms of the endocytic recycling pathway [17], which suggests that Neo1p is also a flippase.

*ANY1/CFS1* was identified as a suppressor mutation of growth defects in the *neo1Δ* and *cdc50Δ* mutants [18], [19]. Cfs1p belongs to the PQ-loop family, which has seven-helix membrane topology, and partially colocalizes with Drs2p and Neo1p. The *cfs1Δ* mutation completely suppressed the lethality of the *neo1Δ* mutant, and the *neo1Δ cfs1Δ* mutant did not exhibit any defects in membrane trafficking. However, the *neo1Δ cfs1Δ* mutant may not be equivalent to the wild type in its Golgi functions, otherwise a set of these two genes including the essential *NEO1* gene would be dispensable. Thus, a number of genes that are functionally relevant to phospholipid asymmetry may function to alleviate defects in the endosomal/Golgi membrane function in the *neo1Δ cfs1Δ* mutant. To determine such genes, we performed synthetic lethal screening with the *neo1Δ cfs1Δ* mutations, and the *ERD1* gene was identified. Erd1p is involved in the transport of phosphate (Pi) from the luminal space of the Golgi membrane to the cytoplasm [20]. Our results suggest that phospholipid asymmetry and luminal Pi concentration coordinately regulate functions of the Golgi membrane.

## Materials and methods

### Media and genetic methods

Chemicals were purchased from Wako Pure Chemicals Industry (Osaka, Japan), unless otherwise stated. Standard genetic manipulations of the yeast were performed according to a previously described method [21]. Yeast transformations were performed using the lithium acetate method [22], [23]. Yeast strains were then cultured in YPDA-rich medium [1% yeast extract (Difco Laboratories, Detroit, MI), 2% Bacto-peptone (Difco), 2% glucose, and 0.01% adenine]. Strains that carry plasmids were placed in a synthetic medium [SD: 0.67% yeast nitrogen base (YNB) without amino acids (Difco) and 2% glucose] that contain required nutritional

supplements [24]. The SDA medium was a SD medium that contains 0.5% casamino acid (Difco). To induce the *GAL1* promoter, 3% galactose and 0.2% sucrose, instead of glucose, were used as carbon sources (YPGA and SG-Leu media). When expression of the *GAL1* promoter was attenuated, 2% raffinose and either 0.01% or 0.1% galactose were used as carbon sources. A Pi-depleted SD medium was prepared with YNB without amino acids and phosphate (ForMedium Ltd., Hunstanton, UK). To adjust the Pi concentration in the SD and YPDA medium at pH 6.6, 1 M $K_2HPO_4$ and 1 M $KH_2PO_4$ were added at a ratio of 38.1: 61.9.

## Yeast strains and plasmids

The yeast strains used in this study are listed in the "Supplemental Material" section (S1 Table). PCR-based procedures were used to construct gene deletions and gene fusions with the *GAL1* promoter, green fluorescent protein (GFP), mCherry, and monomeric red fluorescent protein 1 (mRFP1) [25], [26]. Standard molecular biological techniques were used for plasmid construction, PCR amplification, and DNA sequencing [27]. *Escherichia coli* strain XL1-Blue was used to construct and amplify the plasmids. Gene deletions of *CFS1*, *ERD1*, *BST1*, *EMP24*, *ESP1*, *ERP1*, *ERP2*, *RER1*, *VAN1*, *MNN10*, *MNN2*, *MNN5*, *MNN4*, *MNN6*, and *MNN1* in the YEF 473 genetic background [28] were performed as follows. Regions with the *KanMX6* disruption marker and flanking sequences were PCR-amplified using genomic DNA that were derived from the knockout strain in the BY4741 background [29] as a template. The amplified DNA fragments were introduced into the appropriate strains, and G418-resistant transformants were selected. To construct *KanMX6::P_{PIK1}-GFP-PIK1* strains, the *TRP1::P_{GAL1}-GFP-PIK1* allele was replaced with the *KanMX6::P_{PIK1}-GFP-PIK1* fragment by the PCR-based allele replacement method [30]. All constructs that were produced by the PCR-based procedure were verified by colony PCR amplification to confirm that the replacement or insertion occurred at the expected locus. Sequences of PCR primers are available on request.

The plasmids of this study are listed in the "Supplemental Material" section (S2 Table). To construct pRS316-mCherry-evt-2 PH (pKT2205), the mCherry-evt-2 PH fragment from pmCherry-Evc2-C2 (a gift from T. Taguchi, Tohoku University) was cloned into the *Nhe*I–*Bam*HI gap of pRS316-$P_{TPI1}$-$T_{ADH1}$. To construct pRS306-*2× UPRE-GFP* (pKT2196), the 2× UPRE [31] and GFP fragments were inserted into the *Hind*III–*Kpn*I gap of pRS306-$P_{TPI1}$-$T_{ADH1}$. These pRS306-based plasmids were linearized and integrated into the *URA3* locus. pRS416-$P_{TPI1}$-*GFP-PHO87* (pKT2203) and pRS416-$P_{TPI1}$-*GFP-PHO90* (pKT2204) were constructed by inserting the coding regions of *PHO87* and *PHO90*, respectively, into the *Bam*H1–*Sal*1 gap of pRS416-$P_{TPI1}$-*GFP-PEP12*-$T_{PEP12}$ (pKT1487). Multicopy plasmids that carry *ERD1*, *PHO87*, *PHO90*, *ERS1*, *SAR1*, or *YIP1* were constructed by ligating DNA fragments, which were PCR-amplified using the genomic DNA of wild type cells (YKT38), into the YEplac181 vector plasmid. Schemes that detail the construction of plasmids and DNA sequences of nucleotide primers are available on request.

## Isolation of mutants that are synthetically lethal with the *neo1Δ cfs1Δ* mutation

Mutants that are synthetically lethal with *neo1Δ cfs1Δ* were isolated according to a previously described procedure [32]. From 11,100 colonies that were screened, three single recessive mutations were identified by genetic analyses, and the corresponding wild type genes were cloned. These genes encoded *DRS2*, *CDC50*, and *ERD1*. Mutations in *DRS2*/*CDC50* were shown to be synthetically lethal with *neo1Δ cfs1Δ* [18]. Likewise, the present study confirms that the *erd1Δ* mutation is synthetically lethal with *neo1Δ cfs1Δ* (Fig 1A).

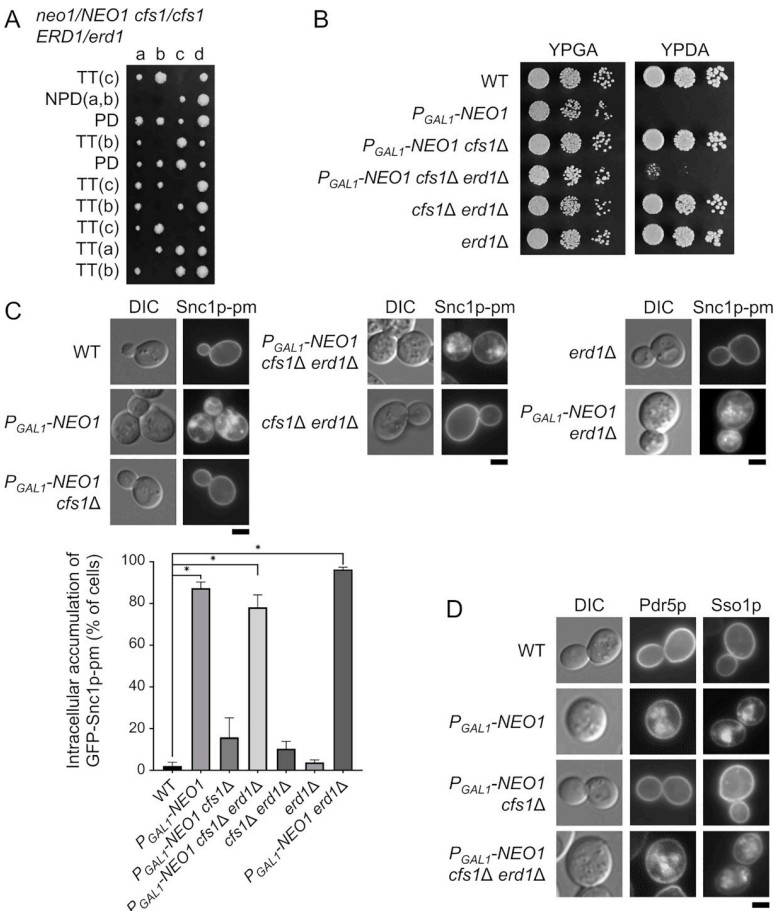

**Fig 1. Synthetic lethality and membrane trafficking defects in the *neo1Δ cfs1Δ erd1Δ* mutant.** (A) The *erd1Δ* mutation exhibited synthetic lethality with the *neo1Δ cfs1Δ* mutations. Tetrad dissection of the *neo1Δ/NEO1 cfs1Δ/ cfs1Δ ERD1/erd1* diploid. Tetrad genotypes (TT, tetra type; PD, parental ditype; NPD, nonparental ditype) are indicated, and the identities of the triple mutant are shown in parentheses. (B) The growth defects of the Neo1p-depleted *cfs1Δ erd1Δ* mutant. Cells were grown to the early log phase in YPDA for approximately 12 h to deplete Neo1p, washed, and adjusted to a concentration of $1.0 \times 10^7$ cells/mL. Next, 4-μl drops of 4-fold serial dilutions were spotted onto YPGA (galactose) and YPDA (glucose) plates and then incubated at 30°C for 1.5 d. The strains used were wild type (WT) (YKT38), $P_{GAL1}$-*NEO1* (YKT2134), $P_{GAL1}$-*NEO1 cfs1Δ* (YKT2135), $P_{GAL1}$-*NEO1 cfs1Δ erd1Δ* (YKT2136), *cfs1Δ erd1Δ* (YKT2137), and *erd1Δ* (YKT2138). (C) Intracellular accumulation of GFP-Snc1p-pm in the Neo1p-depleted *cfs1Δ erd1Δ* mutant. *Upper panels*: Strains that express GFP-Snc1p-pm were grown to the exponential phase in YPDA medium at 30°C for ~20 h except for $P_{GAL1}$-*NEO1* and $P_{GAL1}$-*NEO1 erd1Δ* (12 h) to deplete Neo1p, followed by observation using a fluorescence microscope. The strains used were WT (YKT2139), $P_{GAL1}$-*NEO1* (YKT2140), $P_{GAL1}$-*NEO1 cfs1Δ* (YKT2141), $P_{GAL1}$-*NEO1 cfs1Δ erd1Δ* (YKT2142), *cfs1Δ erd1Δ* (YKT2143), *erd1Δ* (YKT2144), and $P_{GAL1}$-*NEO1 erd1Δ* (YKT2225), all of which carry *GFP-SNC1-pm*. *Lower panel*: The percentage of cells with intracellularly accumulated GFP-Snc1p-pm was determined ($n = 200$) and is shown as the mean ± standard deviation of five independent experiments. Asterisks indicate a significant difference, as determined by the Tukey–Kramer test (*: $p < 0.01$). (D) Intracellular accumulation of Pdr5p-GFP and GFP-Sso1p in the Neo1p-depleted *cfs1Δ erd1Δ* mutant. Strains were cultured and observed as in (C), except that *GFP-SSO1* strains were cultured in SDA-Ura. The strains used were WT (YKT38 and YKT2145), $P_{GAL1}$-*NEO1* (YKT2134 and YKT2146), $P_{GAL1}$-*NEO1 cfs1Δ* (YKT2135 and YKT2147), and $P_{GAL1}$-*NEO1 cfs1Δ erd1Δ* (YKT2148 and YKT2149), all of which carry pRS416-*GFP-SSO1* (pKT1476) and *PDR5-GFP*, respectively. Bar, 5 μm. DIC, differential interference contrast.

## Isolation of multicopy suppressors of the *neo1Δ cfs1Δ erd1Δ* mutant

The *neo1Δ cfs1Δ erd1Δ* strain (YKT2173) with the YCplac33-*NEO1* plasmid (pKT1470) was transformed using a yeast genomic DNA library that was constructed with the multicopy plasmid YEp13 [33]. Transformants were spread onto SD-Leu plates and incubated at 30°C for 3

days. These plates were replica-plated onto SD-Leu containing 0.1% 5-fluoroorotic acid (5-FOA) plates to select clones that had lost *URA3*-containing YCplac33-*NEO1* [34]. Approximately $3 \times 10^5$ transformants were screened, and 190 clones were obtained, of which 61 were further analyzed. 18 clones were found to contain *NEO1* and *ERD1* by colony PCR and were thus eliminated. Plasmids were then recovered from the remaining 43 clones, and restriction enzyme digestion of the plasmids indicated that they originated from 13 independent clones. These 13 plasmids were reintroduced into the original mutant to confirm the suppression. Finally, they were grouped into five different genomic regions by DNA sequencing. Fragment subcloning revealed that *PHO87*, *PHO90*, *ERS1*, *SAR1* and *YIP1* were responsible for the suppression.

## Microscopic observations

Cells were observed using a Nikon ECLIPSE E800 microscope (Nikon Instec, Tokyo, Japan) equipped with an HB-10103AF superhigh-pressure mercury lamp and a 1.4 numerical aperture 100 × Plan Apo oil immersion objective lens (Nikon Instec) with appropriate fluorescence filter sets (Nikon Instec) or differential interference contrast optics. Images were acquired using a cooled digital charge-coupled device camera (C4742-95-12NR; Hamamatsu Photonics, Hamamatsu, Japan) and AQUACOSMOS software (Hamamatsu Photonics). GFP-, mRFP1-, or mCherry-tagged proteins were observed in living cells, which were grown to early to mid-logarithmic phase, harvested, and resuspended in SD medium. Cells were immediately observed using a GFP bandpass (for GFP) or G2-A (for mRFP1 and mCherry) filter set. Observations were compiled from examining at least 200 cells. For statistical analysis, the observation of 200 cells was repeated five times for each strain.

## Results

### Isolation of *erd1Δ* as a mutation that is synthetically lethal with *neo1Δ cfs1Δ* mutations

We previously showed that the *cfs1Δ* mutation suppressed the lethality of the *neo1Δ* mutant. The *cfs1Δ neo1Δ* mutant grew normally, and the *cfs1Δ* mutation efficiently suppressed the membrane trafficking defects of the Neo1p-depleted cells [18]. The *neo1* mutations caused abnormal phospholipid asymmetry; the *neo1* mutants exposed PE and PS in the PM [16]. The *any1Δ/cfs1Δ* mutation completely suppressed PE and PS exposure in the *neo1-2* temperature-sensitive mutant [19], whereas it did not suppress the exposure in the *neo1Δ* mutant [35], which suggests that Cfs1p partially antagonizes the flippase action of Neo1p. To isolate a gene that is functionally relevant to *NEO1* and *CFS1*, we performed synthetic lethal screening using the *neo1Δ cfs1Δ* mutant. Consequently, a mutation in *ERD1* was isolated (Fig 1A). *ERD1* was identified as a gene that encodes a membrane protein required for the retention of proteins that are localized to the endoplasmic reticulum (ER) and was subsequently shown to be required for glycosylation of some proteins in the Golgi apparatus [36], [37]. A recent study suggested that Erd1p transports Pi from the lumen of the Golgi apparatus to the cytoplasm and recycles the Pi byproducts of glycosylation reactions [20]. To phenotypically analyze the *neo1 cfs1 erd1* mutant, we constructed the $P_{GAL1}$-*NEO1 cfs1Δ erd1Δ* mutant, in which the expression of *NEO1* is controlled by the glucose-repressible *GAL1* promoter. The *erd1Δ* mutation exhibited synthetic growth defects with the Neo1p-depleted *cfs1Δ* mutation, but not with the *cfs1Δ* mutation (Fig 1B). We examined whether other mutations that are involved in ER retention exhibited synthetic lethality with the Neo1p-depleted *cfs1Δ* mutation, but none did (S1A Fig). We also examined mutations in Golgi glycosylation, but they also were not

synthetically lethal with Neo1p-depleted *cfs1Δ* (S1B Fig). These results suggest that a defect specific to *erd1Δ* causes synthetic lethality with Neo1p-depleted *cfs1Δ*.

We examined membrane trafficking in the Neo1p-depleted *cfs1Δ erd1Δ* mutant using GFP-Snc1p-pm as a marker protein. Snc1p-pm is a PM-localized mutant *v*-SNARE with point mutations that inhibit endocytosis [38]. Neo1p-depletion intracellularly accumulated GFP-Snc1p-pm, and the *cfs1Δ* mutation suppressed this phenotype, as previously reported [18] (Fig 1C). The Neo1p-depleted *cfs1Δ erd1Δ* mutant accumulated GFP-Snc1p-pm like Neo1p-depleted cells (Fig 1C). We also examined the localization of Pdr5p and Sso1p, which are two GFP-fused PM proteins. Pdr5p is an ATP-binding cassette transporter [39],while Sso1p is a *t*-SNARE that is involved in the fusion of secretory vesicles with the PM [40]. Similar to GFP-Snc1p-pm, Neo1p-depletion accumulated Pdr5p-GFP and GFP-Sso1p in ~77% and 94% of the cells, respectively, (*n* = 200 cells) (Fig 1D). The Neo1p-depleted *cfs1Δ* mutant also accumulated Pdr5p-GFP and GFP-Sso1p to a certain extent (~16% and 28%, respectively, *n* = 200 cells). In contrast, the Neo1p-depleted *cfs1Δ erd1Δ* mutant accumulated Pdr5p-GFP and GFP-Sso1p to a large extent (~45% and 84%, respectively, *n* = 200 cells), although the accumulation of Pdr5p-GFP was relatively low compared to GFP-Snc1p-pm and GFP-Sso1p. These results suggest that the Neo1p-depleted *cfs1Δ erd1Δ* mutant exhibits major defects in membrane trafficking pathways.

## PM proteins accumulate in the TGN in the Neo1p-depleted *cfs1Δ erd1Δ* mutant

Neo1p is localized to the TGN, *cis*- and *medial*-Golgi, and endosomal compartments [14], [15], [41]. Cfs1p is partially localized to the TGN [18], whereas the localization of Erd1p has not yet been determined. Erd1p-GFP, which was expressed from its own promoter, was barely detectable, and thus Erd1p-GFP was overexpressed under the control of the *GAL1* promoter. We compared the localization of Erd1p-GFP to that of Mnn9p-mRFP1, which is a *cis*-Golgi marker [42], and Sec7p-mRFP1, which is a TGN marker [43]. Erd1p-GFP colocalized with Mnn9p-mRFP1, rather than with Sec7p-mRFP1 (Fig 2A); 44% of the Erd1p-GFP dots were colocalized with Mnn9p-mRFP1 dots (*n* = 221), and 43% of the Mnn9p-mRFP1 dots were co-localized with Erd1p-GFP dots (*n* = 225). In contrast, 9% of the Erd1p-GFP dots were colocalized with Sec7p-mRFP1 dots (*n* = 316), and 12% of the Sec7p-mRFP1 dots were colocalized with Erd1p-GFP dots (*n* = 221). These results suggest that Erd1p is mainly localized to the *cis*-Golgi.

The localization patterns of Neo1p, Cfs1p, and Erd1p suggest that the Neo1p-depleted *cfs1Δ erd1Δ* mutant has defective membrane trafficking through the Golgi apparatus. We examined whether GFP-Snc1p-pm was accumulated in the *cis*-Golgi or the TGN in the Neo1p-depleted *cfs1Δ erd1Δ* mutant. In both the Neo1p-depleted and Neo1p-depleted *cfs1Δ erd1Δ* mutants, 25% (*n* = 224) and 26% (*n* = 237) of GFP-Snc1p-pm-positive structures were colocalized with Mnn9p-mCherry structures, respectively (Fig 2B). In contrast, 87% (*n* = 200) and 70% (*n* = 215) of GFP-Snc1p-pm structures were colocalized with Sec7p-mRFP1 structures in the Neo1p-depleted and Neo1p-depleted *cfs1Δ erd1Δ* mutants, respectively (Fig 2C). These results suggest that Snc1p-pm mainly accumulates in the TGN of the Neo1p-depleted *cfs1Δ erd1Δ* mutant.

## Abnormal distribution of phospholipids in the TGN of the Neo1p-depleted *cfs1Δ erd1Δ* mutant

We next examined the distribution of phospholipids in the Neo1p-depleted *cfs1Δ erd1Δ* mutant. The C2 domain of lactadherin (Lact-C2) and the pleckstrin homology (PH) domain

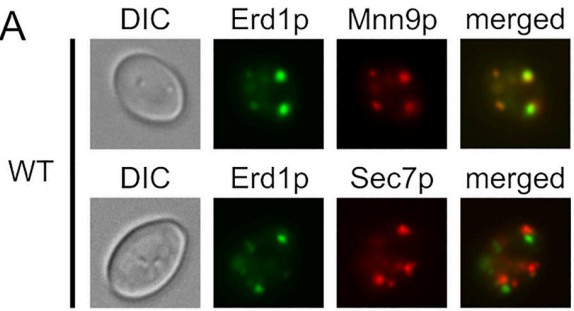

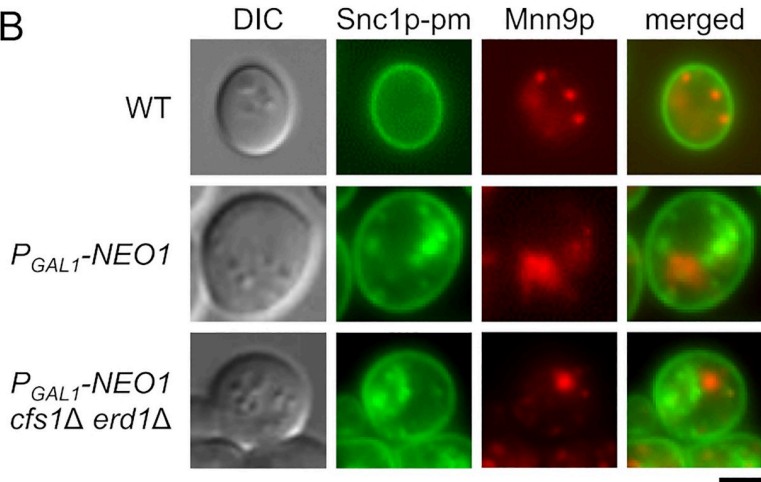

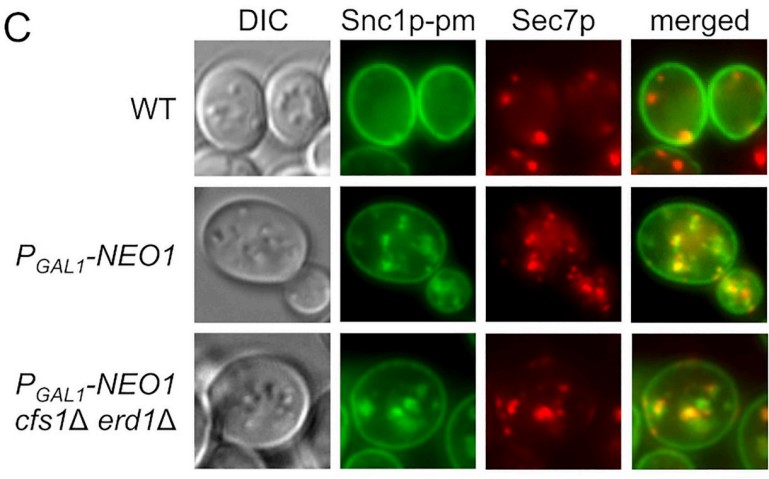

**Fig 2. Localization of GFP-Snc1p-pm in the Neo1p-depleted *cfs1Δ erd1Δ* mutant.** (A) Colocalization of Erd1p-GFP with Mnn9p-mRFP1. Cells were cultured in YPGA at 30˚C for 7 h, followed by microscopic observation. The strains used were $P_{GAL1}$-*ERD1-GFP MNN9-mRFP1* (YKT2150) and $P_{GAL1}$-*ERD1-GFP SEC7-mRFP1* (YKT2151). (B) Localization of GFP-Snc1p-pm and Mnn9p-mCherry in the Neo1p-depleted *cfs1Δ erd1Δ* mutant. Cells were cultured in YPDA at 30˚C as in Fig 1C. The strains used were wild type (WT) (YKT2152), $P_{GAL1}$-*NEO1* (YKT2153), and $P_{GAL1}$-*NEO1 cfs1Δ erd1Δ* (YKT2154), all of which carry *GFP-SNC1-pm* and *MNN9-mCherry*. (C) Localization of GFP-Snc1p-pm and Sec7p-mRFP1 in the Neo1p-depleted *cfs1Δ erd1Δ* mutant. Cells were cultured in YPDA at 30˚C as in Fig 1C.

The strains used were WT (YKT2155), $P_{GAL1}$-NEO1 (YKT2156), and $P_{GAL1}$-NEO1 cfs1Δ erd1Δ (YKT2157), all of which carry *GFP-SNC1-pm* and *SEC7-mRFP1*. Bar, 5 μm. DIC, differential interference contrast.

of evectin-2 (evt-2 PH) were used as probes for PS in the cytoplasmic leaflet of the PM and organelle membranes [44], [45]. GFP-Lact-C2 normally localizes only to the PM, but it localizes to the TGN membranes in flippase mutants, which suggests that PS is exposed to the cytoplasmic leaflet of the TGN [46]. In the $P_{GAL1}$-NEO1 mutant, 67% ($n = 207$) and 71% ($n = 208$) of the GFP-Snc1p-pm structures were colocalized with mRFP1-Lact-C2 and mCherry-evt-2 PH structures, respectively (Fig 3A and 3B). Similarly, in the Neo1p-depleted cfs1Δ erd1Δ mutant, 51% ($n = 255$) and 61% ($n = 206$) of GFP-Snc1p-pm structures were colocalized with mRFP1-Lact-C2 and mCherry-evt-2 PH structures, respectively. These results suggest that PS is exposed to the cytoplasmic leaflet of the TGN in the Neo1p-depleted cfs1Δ erd1Δ mutant.

We next examined the localization of phosphatidylinositol-4-phosphate (PI4P) in the TGN. PI4P, which is produced by the PI4P kinase Pik1p, plays a pivotal role in the vesicle transport from the TGN to the PM and endosomes [47]. A tandem dimer of the PH domain of Osh2p was shown to detect the Golgi pool of PI4P in addition to the PM pool [48]. Osh2p-PH-GFP was mainly localized to Golgi-like structures in wild-type cells but was mainly localized to the PM in the Neo1p-depleted and Neo1p-depleted cfs1Δ erd1Δ mutants (Fig 3C). Cell counting indicated that more than 57% of the cells expressed the Osh2p-PH-GFP signal only at the PM in both the Neo1p-depleted and Neo1p-depleted cfs1Δ erd1Δ mutants. The Neo1p-depleted cfs1Δ mutant also demonstrated mislocalization to a significant extent (~32%). We confirmed that Osh2p-PH-GFP was not localized to the TGN in the Neo1p-depleted and Neo1p-depleted cfs1Δ erd1Δ mutants of Sec7p-mRFP1-expressing cells (Fig 3D). These results suggest that PI4P is absent from the TGN, and this may cause membrane trafficking defects in the Neo1p-depleted and Neo1p-depleted cfs1Δ erd1Δ mutants, which is consistent with our previous finding that Pik1p depletion caused intracellular accumulation of GFP-Snc1p-pm [46]. We then examined the localization of GFP-Pik1p. Interestingly, GFP-Pik1p was normally colocalized with Sec7p-mRFP1 in the Neo1p-depleted and Neo1p-depleted cfs1Δ erd1Δ mutants (Fig 3E). In the Neo1p-depleted and Neo1p-depleted cfs1Δ erd1Δ mutants, 89% ($n = 224$) and 95% ($n = 213$) of GFP-Pik1p structures were colocalized with Sec7p-mRFP1 structures, respectively. These results suggest that either Pik1p is inactivated or limited phosphatidylinositol (PI) is present in the TGN of these mutants. Related to this, the neo1Δ cfs1Δ mutant may have defects in the production of PI4P, as partial depletion of Pik1p inhibited the growth of this mutant (S2 Fig).

## Pi homeostasis in the Golgi lumen is crucial for viability of the Neo1p-depleted cfs1Δ mutant

To gain insight to the synthetic lethality of the neo1Δ cfs1Δ erd1Δ mutant, we performed multicopy suppressor screening. A neo1Δ cfs1Δ erd1Δ strain that contains a *NEO1/URA3* plasmid was transformed with a genomic library, and the transformants were then screened on a plate with 5-fluoroorotic acid (5-FOA), as described in the "Materials and Methods" section. As a result, *PHO87*, *PHO90*, *ERS1*, *SAR1*, and *YIP1* were isolated (Fig 4A). Further, these multicopy suppressors were found to suppress the growth defects of the Neo1p-depleted cfs1Δ erd1Δ mutant (Fig 4B), but not those of the Neo1p-depleted mutant (Fig 4C), which suggests that the multicopy suppressors suppressed the erd1Δ mutation or the synthetic growth defects that are associated with the Neo1p-depleted cfs1Δ erd1Δ mutant. Pho87p and Pho90p are low-affinity Pi transporters at the PM [49], [50], [51]. *ERS1* was isolated as a multicopy suppressor of the

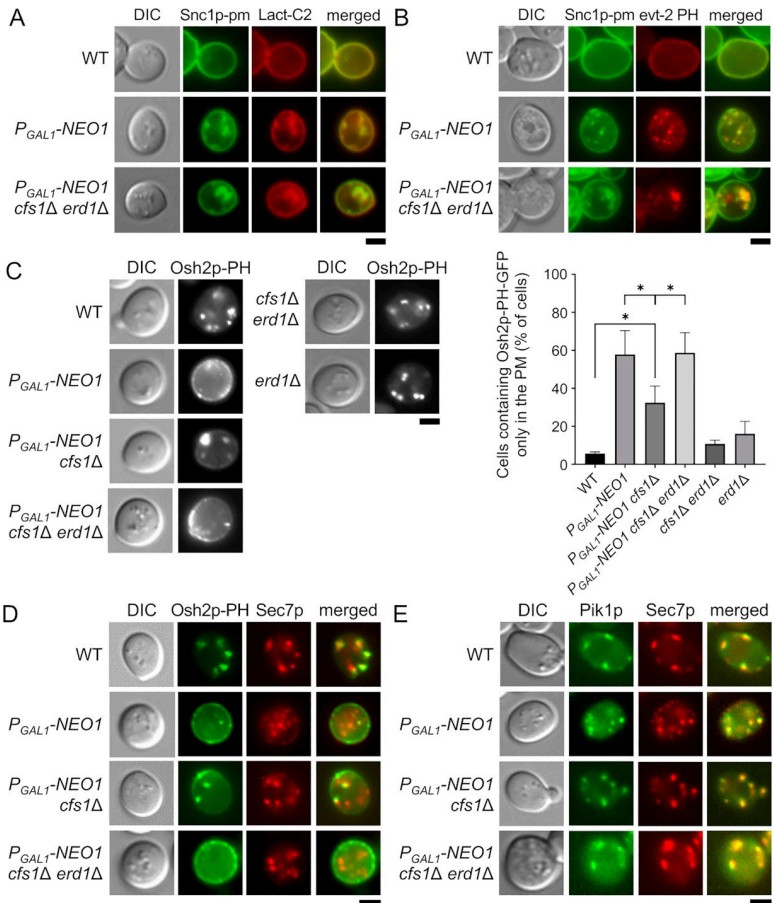

**Fig 3. The cytoplasmic leaflet of the TGN in the Neo1p-depleted *cfs1Δ erd1Δ* mutant contains PS and decreased level of PI4P.** (A) and (B) PS is exposed to the cytoplasmic leaflet of the TGN in the Neo1p-depleted *cfs1Δ erd1Δ* mutant. (A) Localization of mRFP1-Lact-C2. Cells were cultured in YPDA at 30°C as in Fig 1C. The strains used were wild type (WT) (YKT2139), *P_GAL1-NEO1* (YKT2158), and *P_GAL1-NEO1 cfs1Δ erd1Δ* (YKT2142), all of which carry *GFP-SNC1-pm* and pRS416-mRFP1-Lact-C2 (pKT1755). (B) Localization of mCherry-evt-2 PH. Cells were cultured in YPDA at 30°C as in Fig 1C. The strains used were WT (YKT2139), *P_GAL1-NEO1* (YKT2158), and *P_GAL1-NEO1 cfs1Δ erd1Δ* (YKT2142), all of which carry *GFP-SNC1-pm* and pRS316-mCherry-evt-2 PH (pKT2205). (C) and (D) PI4P was absent from the TGN in the Neo1p-depleted *cfs1Δ erd1Δ* mutant. (C) Localization of Osh2p-PH-GFP. *Left panels*: Cells were cultured in YPDA at 30°C as in Fig 1C. The strains used were WT (YKT2159), *P_GAL1-NEO1* (YKT2160), *P_GAL1-NEO1 cfs1Δ* (YKT2161), *P_GAL1-NEO1 cfs1Δ erd1Δ* (YKT2162), *cfs1Δ erd1Δ* (YKT2163), and *erd1Δ* (YKT2164), all of which carry *OSH2-PH-GFP*. *Right panel*: The percentage of cells with Osh2p-PH-GFP only at the PM was determined (*n* = 200) and is expressed as the mean ± standard deviation of five independent experiments. Asterisks indicate a significant difference, as determined by the Tukey–Kramer test (*: p < 0.01). (D) Localization of Osh2p-PH-GFP and Sec7p-mRFP1. Cells were cultured in YPDA at 30°C as in Fig 1C. The strains used were WT (YKT2165), *P_GAL1-NEO1* (YKT2166), *P_GAL1-NEO1 cfs1Δ* (YKT2167), and *P_GAL1-NEO1 cfs1Δ erd1Δ* (YKT2168), all of which carry *OSH2-PH-GFP* and *SEC7-mRFP1*. (E) Colocalization of GFP-Pik1p and Sec7p-mRFP1 in the Neo1p-depleted *cfs1Δ erd1Δ* mutant. Cells were cultured in YPDA at 30°C as in Fig 1C. The strains used were WT (YKT2169), *P_GAL1-NEO1* (YKT2170), *P_GAL1-NEO1 cfs1Δ* (YKT2171), and *P_GAL1-NEO1 cfs1Δ erd1Δ* (YKT2172), all of which carry *P_PIK1-GFP-PIK1* and *SEC7-mRFP1*. Bar, 5 μm. DIC, differential interference contrast.

*erd1Δ* mutant [52] and Ers1p may function as a cystine transporter in vacuoles [53]. Sar1p regulates the formation of COPII vesicles from the ER [54], [55], and Yip1p is also implicated in COPII vesicle biogenesis [56]. We confirmed that all these multicopy suppressors suppressed the intracellular accumulation of GFP-Snc1p-pm and mislocalization of Osh2p-PH-GFP in the Neo1p-depleted *cfs1Δ erd1Δ* mutant (Fig 4D).

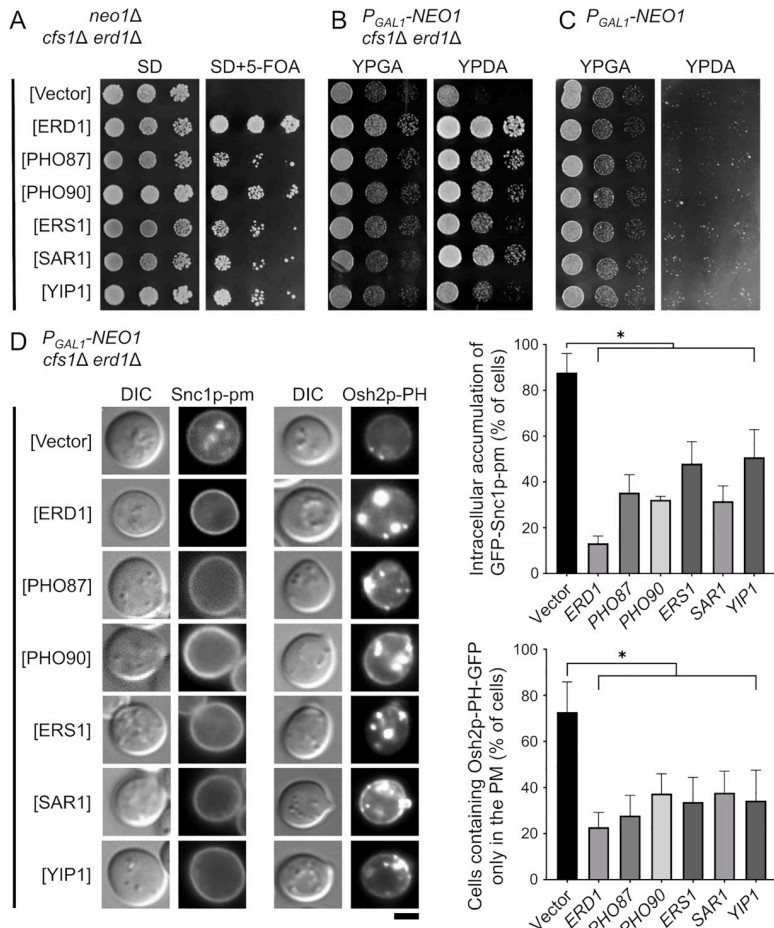

**Fig 4. Isolation of multicopy suppressors in the *neo1Δ cfs1Δ erd1Δ* mutant.** (A) Suppression of the growth defects of the *neo1Δ cfs1Δ erd1Δ* mutant. A spot assay for growth was performed as in Fig 1B; 8-μl drops of 4-fold serial dilutions were spotted onto SD-L and SD-L+5-fluoroorotic acid (FOA) plates and then incubated at 30°C for 4 d. Only cells that lost the plasmid harboring the *NEO1* gene grew on the 5-FOA medium. The strain used was the *neo1Δ cfs1Δ erd1Δ* mutant with YCplac33-*NEO1* (pKT1470) (YKT2173). This strain was transformed with YEplac181(Vector), pKT2197 (ERD1), pKT2198(PHO87), pKT2199(PHO90), pKT2200(ERS1), pKT2201(SAR1), and pKT2202(YIP1). (B) Suppression of the growth defects of the Neo1p-depleted *cfs1Δ erd1Δ* mutant. A spot assay was performed as in (A), except that 4-μl drops were spotted onto YPGA (galactose) and YPDA (glucose) plates and then incubated at 30°C for 1.5 d. The strain used was the *P_{GAL1}-NEO1 cfs1Δ erd1Δ* mutant (YKT2174). (C) Suppression of the growth defects of the Neo1p-depleted mutant. A spot assay was performed as in (B). The strain used was the *P_{GAL1}-NEO1* mutant (YKT2134). (D) Suppression of the intracellular accumulation of GFP-Snc1p-pm and mislocalization of Osh2p-PH-GFP in the Neo1p-depleted *cfs1Δ erd1Δ* mutant. Cells were cultured in YPDA at 30°C for 20 h, followed by microscopic observation. The strain used was the *P_{GAL1}-NEO1 cfs1Δ erd1Δ* mutant with either GFP-Snc1p-pm (YKT2174) or Osh2p-PH-GFP (YKT2162). Each strain was transformed with the plasmids in (A). *Left panel*: Representative cells are shown. *Right panels*: The percentage of cells with intracellular accumulation of GFP-Snc1p-pm or Osh2p-PH-GFP only in the PM was determined (*n* = 200), and the results are expressed as the mean ± standard deviation of five independent experiments. Asterisks indicate a significant difference, as determined by the Tukey–Kramer test (*: p < 0.01). Bar, 5 μm. DIC, differential interference contrast.

Isolation of *PHO87* and *PHO90* is interesting, as Erd1p is proposed to transport Pi from the lumen of the Golgi to the cytoplasm [20]. Since Pho87p and Pho90p are low-affinity Pi transporters at the PM, our results suggest that the synthetic lethality of the *neo1Δ cfs1Δ erd1Δ* mutant may potentially be caused by decreased concentration of cytoplasmic Pi (Fig 5A). This possibility was examined in the following experiments. We first examined whether increased concentrations of extracellular Pi suppressed the growth defect of the Neo1p-depleted *cfs1Δ*

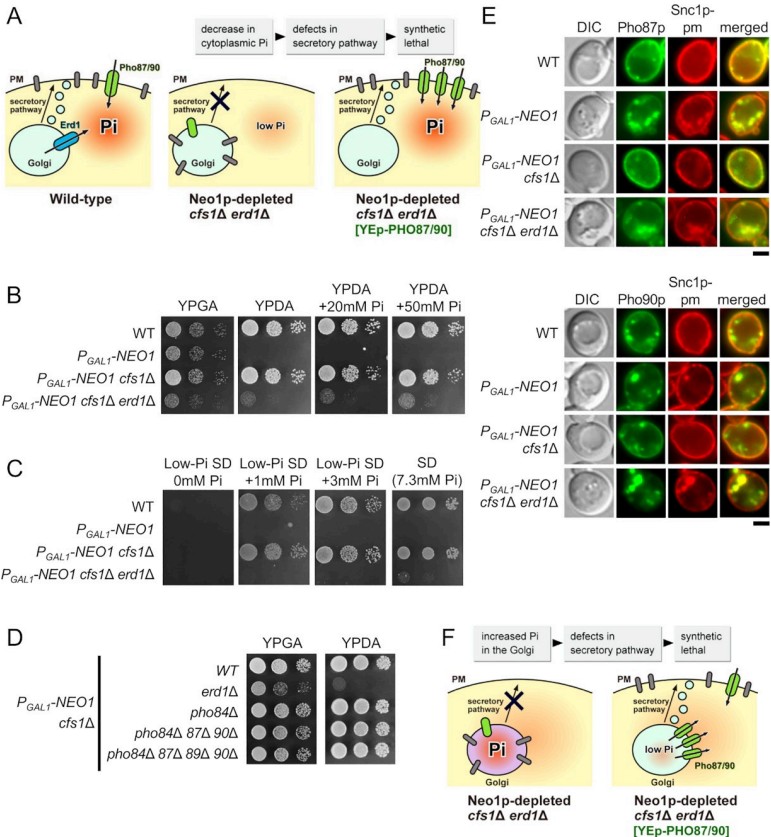

**Fig 5. The synthetic lethality of the *erd1Δ* mutation with the Neo1p-depleted *cfs1Δ* mutation is not caused by reduced levels of cytoplasmic Pi.** (A) The synthetic lethality of *erd1Δ* with Neo1p-depleted *cfs1Δ* may be caused by reduced levels of cytoplasmic Pi. In this model, the overexpression of *PHO87/90* would suppress the growth defect by increasing cytoplasmic Pi. Membrane proteins to be transported to the PM are shown in a grey rod shape. (B) The growth defects of the Neo1p-depleted *cfs1Δ erd1Δ* mutant were not suppressed by increased concentration of extracellular Pi. Cells were grown and spotted, as in Fig 1B, in YPGA (galactose) or YPDA (glucose) medium with either 20 mM Pi or 50 mM Pi and then incubated at 30˚C for 1.5 d. The strains used were wild type (WT) (YKT38), $P_{GAL1}$-*NEO1* (YKT2134), $P_{GAL1}$-*NEO1 cfs1Δ* (YKT2135), and $P_{GAL1}$-*NEO1 cfs1Δ erd1Δ* (YKT2136). (C) The Neo1p-depleted *cfs1Δ* mutant can grow in low-Pi conditions. Cells were grown and spotted as in (B), except that 8-μl drops of 4-fold serial dilutions were spotted onto SD plates with 0, 1, or 3 mM Pi, followed by incubation at 30˚C for 2 d. Standard SD medium contains approximately 7.3 mM Pi according to the manufacturer. The strains used were the same strains as in (B). (D) The Neo1p-depleted *cfs1Δ* mutation is not synthetically lethal with mutations of the Pi transporters at the PM. Cells were grown and spotted as in (B) onto YPGA (galactose) and YPDA (glucose) plates, followed by incubation at 30˚C for 1.5 d. The strains used were $P_{GAL1}$-*NEO1 cfs1Δ* (YKT2135), $P_{GAL1}$-*NEO1 cfs1Δ erd1Δ* (YKT2136), $P_{GAL1}$-*NEO1 cfs1Δ pho84Δ* (YKT2176), $P_{GAL1}$-*NEO1 cfs1Δ pho84Δ pho87Δ pho90Δ* (YKT2177), and $P_{GAL1}$-*NEO1 cfs1Δ pho84Δ pho87Δ pho89Δ pho90Δ* (YKT2178). (E) Localization of GFP-Pho87p and GFP-Pho90p in the Neo1p-depleted *cfs1Δ erd1Δ* mutant. Cells were cultured in YPDA at 30˚C as in Fig 1C. The strains used were wild type (WT) (YKT2179), $P_{GAL1}$-*NEO1* (YKT2180), $P_{GAL1}$-*NEO1 cfs1Δ* (YKT2181), and $P_{GAL1}$-*NEO1 cfs1Δ erd1Δ* (YKT2182), all of which carry *mRFP1-SNC1-pm* and pRS416 $P_{TPI1}$-*GFP-PHO87* (pKT2203) (upper panel) or *mRFP1-SNC1-pm* and pRS416 $P_{TPI1}$-*GFP-PHO90* (pKT2204) (lower panel). Bar, 5 μm. DIC, differential interference contrast. (F) The synthetic lethality of *erd1Δ* with Neo1p-depleted *cfs1Δ* may be caused by increased levels of luminal Pi in the Golgi. In this model, the overexpression of *PHO87/90* would suppress the growth defect by decreasing luminal Pi. Our data seem to be consistent with this model.

*erd1Δ* mutant. The YPD medium contained approximately 5 mM Pi [57]. However, the Neo1p-depleted *cfs1Δ erd1Δ* mutant did not grow in the YPDA medium with either 20 or 50 mM Pi (Fig 5B). If the *erd1Δ* mutation decreased cytoplasmic Pi, and if this caused the synthetic lethality with the Neo1p-depleted *cfs1Δ* mutation, the Neo1p-depleted *cfs1Δ* mutant would express sensitivity to conditions or mutations that decrease cytoplasmic Pi. However,

the Neo1p-depleted *cfs1Δ* mutant grew in a low-Pi medium (1 and 3 mM) (Fig 5C). We next introduced mutations into the Pi transporters of the Neo1p-depleted *cfs1Δ* mutant. In addition to Pho87p and Pho90p, two more Pi transporters reside at the PM, namely Pho84p and Pho89p, which are high-affinity transporters [58], [59]. Even the Neo1p-depleted *cfs1Δ* mutant with mutations in all four transporters grew normally (Fig 5D). These results suggest that increased Pi in the Golgi lumen, rather than decreased Pi in the cytoplasm, may be responsible for the synthetic lethality of the *erd1Δ* mutation with the Neo1p-depleted *cfs1Δ* mutation, and this possibility was examined in subsequent experiments.

Considering that PM proteins accumulated in the TGN of the Neo1p-depleted *cfs1Δ erd1Δ* mutant (Figs 1 and 2), Pho87p and Pho90p may also accumulate in the TGN. We then examined the localization of GFP-Pho87p and GFP-Pho90p, which are expressed under the control of the *TPI1* promoter. In wild-type cells, GFP-Pho87p and GFP-Pho90p were mainly localized to the plasma membrane, but dotty structures were observed. mRFP1-Snc1-pm was exclusively localized to the plasma membrane. In the $P_{GAL1}$-*NEO1* mutant, GFP-Pho87p (85%, $n = 200$) and GFP-Pho90p (82%, $n = 200$) were localized to regions of intracellular accumulation of mRFP1-Snc1-pm (Fig 5E). Similarly, in the Neo1p-depleted *cfs1Δ erd1Δ* mutant, GFP-Pho87p (79%, $n = 200$) and GFP-Pho90p (84%, $n = 200$) were also colocalized with mRFP1-Snc1-pm (Fig 5E). These results suggest that Pho87p and Pho90p are accumulated in and localized to the TGN of the Neo1p-depleted *cfs1Δ erd1Δ* mutant.

Overall, our results suggest that the overexpression of Pho87p or Pho90p accelerates transport of the luminally accumulated Pi to the cytoplasm, which results in the suppression of the growth defects of the *neo1Δ cfs1Δ erd1Δ* mutant (Fig 5F). We propose that the elevated Pi levels of the Golgi lumen are responsible for the lethality of the Neo1p-depleted *cfs1Δ erd1Δ* mutant.

## The Neo1p-depleted *cfs1Δ erd1Δ* mutant also showed defects in the ER structure

Isolation of *SAR1* and *YIP1* as multicopy suppressors led us to examine phenotypes of the ER. To examine the morphology of the ER, Hmg1p-GFP was expressed. Hmg1p is a hydroxy-methylglutaryl-CoA (HMG-CoA) reductase, which catalyzes the conversion of HMG-CoA to mevalonate [60]. Hmg1p localizes to the perinuclear ER [61]. However, its localization was disorganized in the Neo1p-depleted and Neo1p-depleted *cfs1Δ erd1Δ* mutants (Fig 6A). The perinuclear localization of Hmg1p-GFP was absent, and some cells exhibited string-like structures in their cytoplasm. We also examined the nuclear membrane by observing Nup188p, which is a nucleoporin of the nuclear pore complex [62]. Similar to Hmg1p-GFP, nuclear localization of Nup188p-mRFP1 was absent in the Neo1p-depleted and Neo1p-depleted *cfs1Δ erd1Δ* mutants (Fig 6B). We next examined the localization of the cortical ER marker protein Rtn1p, which is a member of the reticulon family and is involved in shaping ER tubules [63]. Interestingly, Neo1p-depleted (78%, $n = 200$) and Neo1p-depleted *cfs1Δ erd1Δ* (29%, $n = 200$) mutants exhibited abnormal dotty aggregates of Rtn1p-GFP (Fig 6C), which was also observed in the *cho2Δ* mutant that is defective in PC biosynthesis [64]. These structures were not observed in either wild-type or Neo1p-depleted *cfs1Δ* cells. These results indicate that the Neo1p-depleted and Neo1p-depleted *cfs1Δ erd1Δ* mutants have structural defects in the ER. However, vesicle transport from the ER was not severely affected, since mRFP1-Snc1-pm did not accumulate in the Hmg1p-GFP structure of these mutants (< 1%, 200 cells) (Fig 6D).

Erd1p was originally isolated as a mutant that secretes a luminal ER protein, such as the ER chaperone Kar2p/BiP [36], [37]. Thus, the *erd1Δ* mutant is defective in the retrieval of Kar2p from the *cis*-Golgi region to the ER. The *neo1* temperature-sensitive mutant also secretes Kar2p and mislocalizes Rer1p [14], which is a retrieval receptor for ER membrane proteins

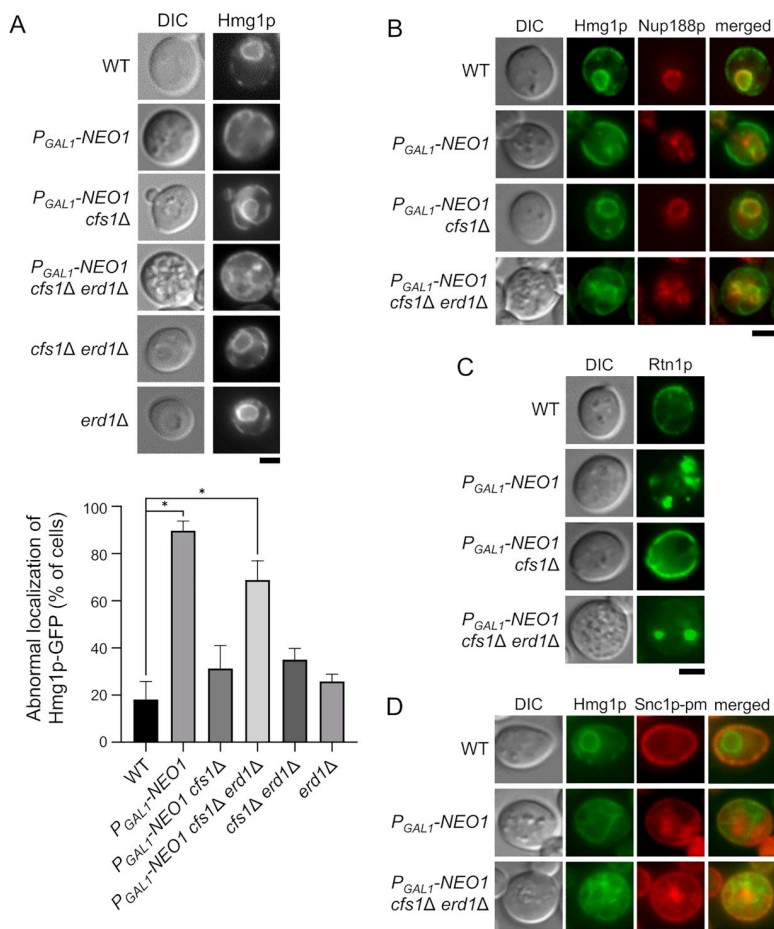

**Fig 6. Disorganization of the ER structure in the Neo1p-depleted *cfs1Δ erd1Δ* mutant.** (A) Localization of Hmg1p-GFP. *Upper panel*: Strains expressing Hmg1p-GFP were cultured in YPDA as in Fig 1C. The strains used were wild type (WT) (YKT2183), *P_GAL1_-NEO1* (YKT2184), *P_GAL1_-NEO1 cfs1Δ* (YKT2185), *P_GAL1_-NEO1 cfs1Δ erd1Δ* (YKT2186), *cfs1Δ erd1Δ* (YKT2187), and *erd1Δ* (YKT2188), all of which carry *HMG1-GFP*. *Lower panel*: The percentage of cells that lacked perinuclear localization of Hmg1p-GFP was determined (*n* = 200) and is expressed as the mean ± standard deviation of five independent experiments. Asterisks indicate a significant difference, as determined by the Tukey–Kramer test (*: p < 0.01). (B) Localization of Nup188p-mRFP1. Cells were cultured in YPDA as in Fig 1C. The strains used were WT (YKT2189), *P_GAL1_-NEO1* (YKT2190), *P_GAL1_-NEO1 cfs1Δ* (YKT2191), and *P_GAL1_-NEO1 cfs1Δ erd1Δ* (YKT2192), all of which carry *HMG1-GFP* and *NUP188-mRFP1*. (C) Localization of Rtn1p-GFP. Cells were cultured in YPDA as in Fig 1C. The strains used were WT (YKT2193), *P_GAL1_-NEO1* (YKT2194), *P_GAL1_-NEO1 cfs1Δ* (YKT2195), and *P_GAL1_-NEO1 cfs1Δ erd1Δ* (YKT2196), all of which carry *RTN1-GFP*. (D) GFP-Snc1p-pm was not colocalized with Hmg1p-GFP in the Neo1p-depleted *cfs1Δ erd1Δ* mutant. Cells were cultured in YPDA as in Fig 1C. The strains used were WT (YKT2197), *P_GAL1_-NEO1* (YKT2198), and *P_GAL1_-NEO1 cfs1Δ erd1Δ* (YKT2199), all of which carry *HMG1-GFP* and *mRFP1-SNC1-pm*. Bar, 5 μm. DIC, differential interference contrast.

[65]. Failure to retain Kar2p in the ER accumulates unfolding proteins in the ER, which induces the unfolded protein response (UPR) [66], [67]. We examined UPR in the Neo1p-depleted *cfs1Δ erd1Δ* mutant with UPR elements (UPRE) that are fused to GFP [68]. Treatment with a reducing agent, such as dithiothreitol (DTT), accumulates unfolding proteins and induces UPRE-GFP expression. The Neo1p-depleted and Neo1p-depleted *cfs1Δ erd1Δ* mutants exhibited strong induction of UPR (Fig 7A). The *erd1Δ* mutation caused weak induction, as previously reported [69]. *ERD2* encodes a receptor for retrieval of luminal ER proteins from the *cis*-Golgi region [70]. The Erd2p-depleted mutant exhibited a strong induction of UPR and the abnormal structure of the ER, which is reminiscent of the Neo1p-depleted *cfs1Δ*

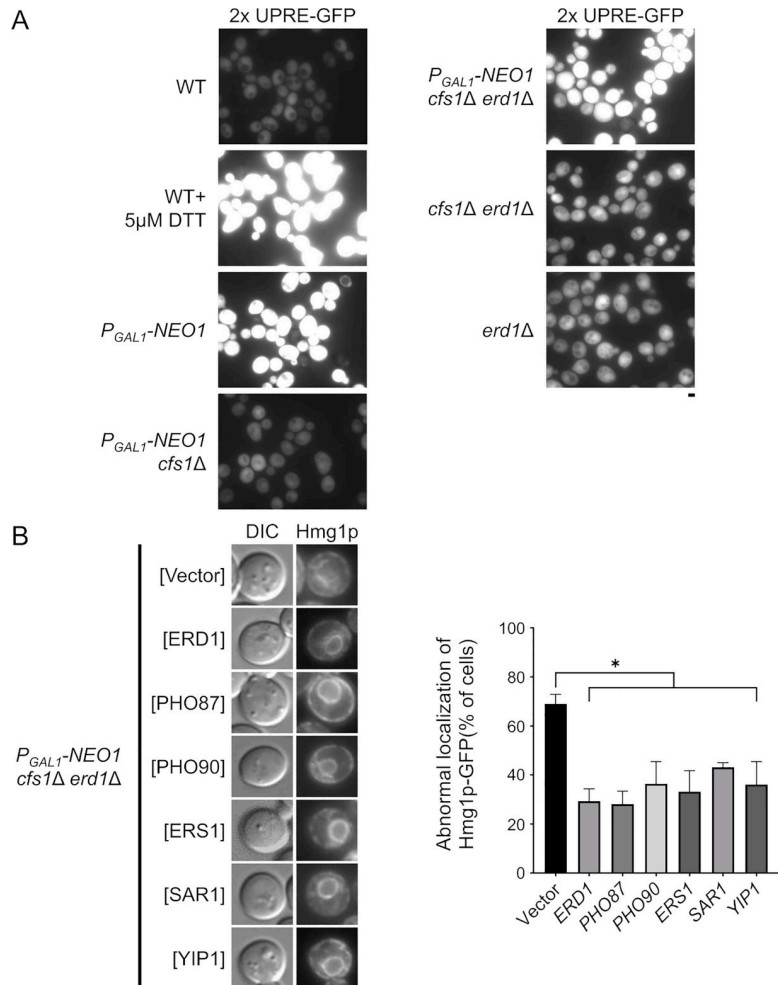

**Fig 7. UPR is induced in the Neo1p-depleted *cfs1Δ erd1Δ* mutant.** (A) Induction of UPR in the Neo1p-depleted
*cfs1Δ erd1Δ* mutant. Yeast cells that express GFP under the control of 2× UPRE were cultured in YPDA as in Fig 1C.
For treatment of wild-type cells with dithiothreitol (DTT), early log phase cells equivalent to 1.5 ml culture of $OD_{600nm}$
= 1.0 were collected and incubated in YPDA that contains 5 μM DTT for 4 h. The strains used were WT (YKT2200),
$P_{GAL1}$-*NEO1* (YKT2201), $P_{GAL1}$-*NEO1 cfs1Δ* (YKT2202), $P_{GAL1}$-*NEO1 cfs1Δ erd1Δ* (YKT2203), *cfs1Δ erd1Δ*
(YKT2204), and *erd1Δ* (YKT2205), all of which carry *2× UPRE-GFP*. (B) Suppression of the abnormal localization of
Hmg1p-GFP by the multicopy suppressors of the Neo1p-depleted *cfs1Δ erd1Δ* mutant. Cells were cultured in YPDA at
30°C for 20 h. The strain used was the $P_{GAL1}$-*NEO1 cfs1Δ erd1Δ* (YKT2186) mutant, which contains *HMG1-GFP* and
multicopy suppressors, as in Fig 4B. *Left panel*: Representative cells are shown. *Right panel*: The percentage of cells with
abnormal localization of Hmg1p-GFP was determined (*n* = 200) and is expressed as the mean ± standard deviation of
five independent experiments. An asterisk indicates a significant difference, as determined by the Tukey–Kramer test
(*: p < 0.01). Bar, 5 μm. DIC, differential interference contrast.

*erd1Δ* mutant, but it exhibited normal transport of Snc1p-pm to the PM (S3 Fig). These morphological and functional defects in the ER of the Neo1p-depleted *cfs1Δ erd1Δ* mutant may be
caused by defective membrane trafficking from the *cis*-Golgi to the ER, similar to the case with
the *neo1* mutants [14].

　We also examined whether the multicopy suppressors suppressed the morphological
defects of the ER in the Neo1p-depleted *cfs1Δ erd1Δ* mutant. All the isolated suppressors suppressed the abnormal structure of the ER (Fig 7B). The mechanism of suppression by *SAR1*
and *YIP1* is unknown, but continued transport of membrane proteins, including Ers1p,
Pho87p, and Pho90p, from the ER to the Golgi may result in the suppression. Alternatively,

enhanced vesicle transport from the ER increases the volume of the Golgi apparatus in the Neo1p-depleted *cfs1Δ erd1Δ* mutant, as membrane transport from the Golgi is inhibited in this mutant. This may dilute Pi in the Golgi lumen, which results in the suppression of membrane trafficking defects through the Golgi apparatus.

## Discussion

In this study, we isolated *erd1* as a mutation that is synthetically lethal with *neo1Δ cfs1Δ* mutations. The Neo1p-depleted *cfs1Δ erd1Δ* mutant exhibited severe defects in Golgi functions, including the anterograde vesicle transport from the TGN and the retrograde transport from the *cis*-Golgi region. Our results suggest that Erd1p is a new factor that is functionally relevant to phospholipid asymmetry. Although flippase activity has not been demonstrated for Neo1p, *neo1* temperature-sensitive mutants exhibited loss of PE asymmetry in the plasma membrane [16]. Likewise, genetic studies demonstrated that Neo1p is functionally relevant to Drs2p [11], [17], [35], [71], which has been most extensively characterized as a flippase [72]. Thus, Cfs1p seems to function in an antagonistic manner to flippases; Cfs1p may be a scramblase or positive regulator of a floppase. Alternatively, Cfs1p may segregate the Neo1p and Drs2p functions, as Drs2p may replace Neo1p in the *neo1Δ cfs1Δ* mutant [35]. In any case, transbilayer phospholipid distribution in the Golgi membrane may not be normally regulated in the *neo1Δ cfs1Δ* mutant.

The role of flippases in membrane trafficking has been extensively studied due to its possible involvement in vesicle formation [8], [9]. The *DRS2/CDC50* complex genetically and physically interacts with the Arf1 small GTPase and its regulators [73], [74], [75]. The *drs2Δ* mutant is defective in the formation of clathrin-coated vesicles of the TGN [73], [76], [77]. Based on the lipid transport activity to the cytoplasmic leaflet, two models have been proposed for the function of flippase in the vesicle formation process: (1) local phospholipid flipping induces membrane curvature, which promotes vesicle formation, and (2) transported lipids (e.g., PS) recruit components, such as adaptors and coat proteins, to facilitate vesicle formation [9], [17], [78]. However, both models are yet to be proven at the molecular level.

Erd1p is required for transport of Pi from the Golgi lumen to the cytoplasm [20]. Pi is a product of the glycosylation reaction, and Pi accumulation in the Golgi lumen seems to be responsible for impaired protein glycosylation and defective retrieval of Kar2p/Bip from the *cis*-Golgi compartment of the *erd1Δ* mutant. Our results suggest that Pi accumulation in the Golgi lumen is responsible for the synthetic lethality of *erd1Δ* with *neo1Δ cfs1Δ* mutations. The combination of abnormal regulation of phospholipid asymmetry and defects of Pi homeostasis leads to severe defects of the vesicle transport process from Golgi membranes. The transbilayer phospholipid distribution in the Golgi membrane is unknown, but asymmetric phospholipid distribution may regulate the function and activity of membrane proteins, including glycosyltransferases, ion transporters, and cargo receptors. Thus, flippase-regulated phospholipid redistribution and luminal Pi concentration seem to coordinately regulate Golgi membrane functions. This mechanism may differ substantially from the direct involvement of flippases in the vesicle formation process. The functional relevance between luminal Pi and phospholipid asymmetry is unknown, but increased Pi alters the ionic environment, which may affect protein-lipid interactions at the luminal surface of the Golgi membrane. Thus, simultaneous changes of the ionic and lipid environments appear to drastically affect membrane trafficking in the Golgi apparatus.

PM proteins were not transported out of the TGN in the Neo1p-depleted and Neo1p-depleted *cfs1Δ erd1Δ* mutants, which may be due to the lack of PI4P at the TGN. Interestingly, the PI4 kinase Pik1p was normally colocalized with Sec7p, which is a binding partner of Pik1p

[79]. Thus, Pik1p is either not activated or PI, which is its substrate, is not sufficient in the cytoplasmic leaflet of the Golgi membrane. PI is also used as a donor of inositol phosphate for the synthesis of complex sphingolipids, inositol phosphoceramide (IPC) and mannosyl-di-inositol phosphoceramide (MIP$_2$C), which is a process that is thought to occur in the luminal leaflet of the Golgi membrane [80]. Therefore, regulation of the transbilayer distribution of PI is an important factor to facilitate Golgi functions. However, PI has not been examined in flippase assays, as a fluorescence-labeled PI is not yet available. PI may be mainly distributed in the luminal leaflet of the Golgi membrane in the Neo1p-depleted and Neo1p-depleted *cfs1Δ erd1Δ* mutants. Regarding future research, an examination of the potential involvement of Neo1p in flipping PI is of particular interest. We previously showed that inositol depletion from the growth medium suppressed the growth defects of Cdc50p-depleted *lem3Δ crf1Δ* and Neo1p-depleted mutants, but not those of the Cdc50p- and Neo1p-depleted mutants [81]. Interestingly, this suppression pattern was the same for the *cfs1Δ* mutation [18]. The suppression mechanism by inositol depletion remains unknown, but it may be relevant to the metabolism or transbilayer distribution of PI, and Cfs1p may also be involved in this process.

## Supporting information

**S1 Fig. Synthetic lethality with the Neo1p-depleted *cfs1Δ* mutation is specific to *erd1Δ*.** (A) Neo1p-depleted *cfs1Δ* is not synthetically lethal with other mutations involved in ER retention. Cells were spotted onto YPGA (galactose) and YPDA (glucose) plates and grown as described in Fig 1B. The strains used were $P_{GAL1}$-*NEO1 cfs1Δ* (WT) (YKT2085) and $P_{GAL1}$-*NEO1 cfs1Δ* carrying *erd1Δ* (YKT2136), *bst1Δ* (YKT2206), *emp24Δ* (YKT2207), *eps1Δ* (YKT2208), *erp1Δ* (YKT2209), *erp2Δ* (YKT2210), and *rer1Δ* (YKT2211). (B) Neo1p-depleted *cfs1Δ* is not synthetically lethal with mutations involved in Golgi glycosylation. Cells were spotted onto YPGA (galactose) and YPDA (glucose) plates and grown as described in (A). The strains used were $P_{GAL1}$-*NEO1 cfs1Δ* (WT) (YKT2085) and $P_{GAL1}$-*NEO1 cfs1Δ* carrying *erd1Δ* (YKT2136), *van1Δ* (YKT2212), *mnn10Δ* (YKT2213), *mnn2Δ* (YKT2214), *mnn5Δ* (YKT2215), *mnn4Δ* (YKT2216), *mnn6Δ* (YKT2217), and *mnn1Δ* (YKT2218). (TIF)

**S2 Fig. Partial depletion of Pik1p impairs growth of the *neo1Δ cfs1Δ* mutant.** Cells were grown and spotted as in Fig 4A onto synthetic medium containing 2% glucose (Gal 0%) or 2% raffinose and 0.01% (Gal 0.01%) or 0.1% (Gal 0.1%) galactose, followed by incubation at 30˚C for 2 d. The strains used were wild type (WT) (YKT38), $P_{GAL1}$-*PIK1* (YKT2219), $P_{GAL1}$-*PIK1 cfs1Δ* (YKT2220), and $P_{GAL1}$-*PIK1 neo1Δ cfs1Δ* (YKT2221). (TIF)

**S3 Fig. Induction of UPR and abnormal structure of the ER in the Erd2p-depleted mutant.** (A) The growth defect of the Erd2p-depleted mutant. Cells were grown and spotted as in Fig 1B onto YPGA (galactose) and YPDA (glucose) plates, followed by incubation at 30˚C for 1.5 d. The strains used were wild type (WT) (YKT38) and $P_{GAL1}$-*ERD2* (YKT2222). (B) Induction of UPR in the Erd2p-depleted mutant. Yeast cells that express GFP under the control of 2× UPRE were cultured in YPDA as in Fig 1C. The strain used was the $P_{GAL1}$-*ERD2* (YKT2222), which carries *2× UPRE-GFP*. (C) Localization of Hmg1p-GFP and Snc1-pm in the Erd2p-depleted mutant. Cells were cultured in YPDA as in Fig 1C. The strains used were the $P_{GAL1}$-*ERD2* with *HMG1-GFP* and *mRFP1-SNC1-pm* (YKT2223) or *GFP-SNC1-pm* and *SEC7-mRFP1* (YKT2224). Bar, 5 μm. DIC, differential interference contrast. (TIF)

**S1 Table.** *Saccharomyces cerevisiae* **strains used in this study.**
(PDF)

**S2 Table. Plasmids used in this study.**
(PDF)

# Acknowledgments

We thank Junpei Ohishi for his contributions to the initial stages of this work. We thank Dr. Tomohiko Taguchi (Tohoku University) for providing the mCherry-evt-2 PH plasmid.

# Author Contributions

**Conceptualization:** Tetsuo Mioka, Kazuma Tanaka.

**Data curation:** Mamoru Miyasaka, Tetsuo Mioka.

**Formal analysis:** Mamoru Miyasaka, Tetsuo Mioka.

**Funding acquisition:** Tetsuo Mioka, Takuma Kishimoto, Kazuma Tanaka.

**Investigation:** Mamoru Miyasaka, Tetsuo Mioka, Takuma Kishimoto, Eriko Itoh.

**Methodology:** Tetsuo Mioka, Takuma Kishimoto.

**Project administration:** Tetsuo Mioka, Kazuma Tanaka.

**Supervision:** Kazuma Tanaka.

**Writing – original draft:** Mamoru Miyasaka, Kazuma Tanaka.

**Writing – review & editing:** Kazuma Tanaka.

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
