## [Decision Letter · Decision Letter 0]

16 Apr 2020

PONE-D-20-06448

A complex genetic interaction implicates that phospholipid asymmetry and phosphate homeostasis regulate Golgi functions

PLOS ONE

Dear Dr. Tanaka,

Thank you for submitting your manuscript to PLOS ONE. I am very sorry for the late response due to the difficulties in obtaining the comments from the reviewers amid the COVID-19 pandemic . After careful consideration, we feel that it has merit but does not fully meet PLOS ONE’s publication criteria as it currently stands. Therefore, we invite you to submit a revised version of the manuscript that addresses the points raised during the review process.

  As you will see from their comments, although both reviewers are very impressed by the findings of an unexpected functional link between phosphate homeostasis and phospholipid transport relevant for a proper functioning of the Golgi, they are not fully convinced by your statements or interpretations of the data.

We would appreciate receiving your revised manuscript by the end of June. To enhance the reproducibility of your results, we recommend that if applicable you deposit your laboratory protocols in protocols.io, where a protocol can be assigned its own identifier (DOI) such that it can be cited independently in the future. For instructions see: http://journals.plos.org/plosone/s/submission-guidelines#loc-laboratory-protocols

We look forward to receiving your revised manuscript.

Kind regards,

Reiko Sugiura, M.D., PhD.

Academic Editor

PLOS ONE

Reviewers' comments:

Reviewer's Responses to Questions

**Comments to the Author**

1. Is the manuscript technically sound, and do the data support the conclusions?

Reviewer #1: No

Reviewer #2: Yes

2. Has the statistical analysis been performed appropriately and rigorously? 

Reviewer #1: Yes

Reviewer #2: Yes

3. Have the authors made all data underlying the findings in their manuscript fully available?

Reviewer #1: Yes

Reviewer #2: Yes

4. Is the manuscript presented in an intelligible fashion and written in standard English?

Reviewer #1: No

Reviewer #2: Yes

5. Review Comments to the Author

Reviewer #1: This study reports on the genetic interactions between a putative flippase, Neo1p, its antagonistic regulator Cfs1p, and a Golgi-resident candidate transporter of inorganic phosphate (Pi), Erd1p. The authors present evidence that cells lacking Neo1p and Cfs1p are particularly sensitive to imbalances in Golgi Pi homeostasis, causing defects in cargo trafficking from the Golgi, an aberrant subcellular distribution of PS and PI4P, and induction of an unfolded protein response. While the study does not provide a coherent mechanistic basis for these observations, the experimental data are of good quality and point at an unexpected link between phosphate homeostasis and phospholipid transport relevant for a proper functioning of the Golgi. However, the manuscript contains some verbose or fuzzy statements and not all conclusions are justified by the experimental data.

COMMENTS

1) I have trouble grasping the meaning of some key statements in the manuscript. For instance, in the Abstract and on p. 4 (bottom) and p. 26 (middle), the authors state “our results suggest that flippase-mediated phospholipid redistribution is functionally vital not only in the cytoplasmic leaflet but also in the luminal leaflet of the Golgi membrane”. I do not see the added value of such a cryptic statement. The study also does not provide any concrete data on the lipid composition of the luminal leaflet in the Golgi of wildtype and mutant cells. I urge the authors to formulate a less ambiguous take-home message, one that focuses more on their principal experimental findings.

2) On p. 19 (bottom) and p. 20 (top), the authors state that their findings demonstrate that overexpression of Pho87p and Pho90p accelerates transport of luminally accumulated Pi to the cytoplasm, and conclude that elevated Pi levels in the Golgi lumen are responsible for the lethality of the Neo1p-depleted csf1/erd1 mutant cells. However, their experimental data provide only indirect evidence for this. Therefore, they should significantly tune down these claims, for instance by replacing the words “demonstrate” for “suggest” and “conclude” for “propose”.

3) To dissect the complex genetic interactions that implicate phospholipid asymmetry and phosphate homeostasis in regulating Golgi function, the authors analyzed the consequences of Neo1p depletion on Golgi function in wildtype, csf1 and csf1/erd1 mutant cells. As Erd1p has been suggested to transport Pi from the lumen of the Golgi, analyzing also the consequences of Neo1p depletion in csf1 mutant cells on Golgi functions would have been particularly informative. However, such experiments are not part of the present study. Why not?

4) p. 14 (top) “…the TGN is exposed to the accumulated PS” should read “…PS is accumulated in the cytosolic leaflet of the TGN”

5) p. 14 (end 1st para) “…the cytoplasmic leaflet of the TGN…. is exposed to PS” should be reformulated as indicated above.

6) Fig. 3 caption “The TGN… is exposed to PS and has no PI4P” should read “The cytoplasmic leaflet of the TGN … contains PS and has no PI4P”

Reviewer #2: Neo1 is a P-type ATPase from the P4 subfamily (P4-ATPase), most members of this subfamily being lipid flippases, i.e. they catalyse lipid transport from the exoplasmic to cytoplasmic leaflet of eukaryotic cell membranes. This activity turns out to be critical for numerous cell functions, ranging from establishment of cell polarity to the regulation of membrane trafficking events. However, the function of Neo1 remains elusive probably partly due to the fact that Neo1 is the only essential P4-ATPase in yeast. The authors previously identified Cfs1, a member of the PQ-loop protein family, as a suppressor of the neo1Δ growth defect. In the present manuscript, to gain further insights into the relationships between Neo1 and Cfs1, Miyasaka and colleagues report on the identification of the erd1 mutation as synthetically lethal with neo1Δcfs1Δ. Erd1 is believed to transport phosphate from the lumen of the Golgi apparatus to the cytosol. Miyasaka and colleagues characterized the neo1Δcfs1Δerd1Δ mutant. Interestingly, the neo1Δcfs1Δerd1Δ and neo1Δ mutants exhibited PI4P defects at the TGN, possibly explaining the trafficking defects of these mutants. The authors further identified genes that suppress the neo1Δcfs1Δerd1Δ phenotype, but not the neo1Δ phenotype, suggesting they suppress the erd1 mutation. Given the identity of the identified suppressors, this work provides a possible mechanism for the synthetic lethality of the Neo1-depleted cfs1Δerd1Δ mutant, namely elevated Pi Golgi levels. Indeed, some of the suppressor genes identified belong to a conserved family of plasma membrane transporters involved in phosphate uptake. Additionally, some of the identified suppressors are proteins involved in the ER to Golgi transport, in line with the ER structural defects observed for the Neo1-depleted cfs1Δerd1Δ mutant and for the neo1 mutant.

Overall, this work proposes a connection between phospholipid asymmetry and luminal Pi levels and significantly contributes to the field. The approach is smart and elegant, the work is very neat, and the manuscript well organized.

My specific comments are as follows:

- In the abstract (lines 32-34) and in the discussion (lines 488-489), the authors propose that phospholipid redistribution is not only important in the cytosolic leaflet but also in the luminal leaflet. Do the authors mean that phospholipid asymmetry has an important impact not only on the cytosolic face of the membrane but also on the luminal one? I find the word ‘leaflet’ misleading in this case, as it restricts the impact of phospholipid asymmetry to the membrane itself

- Introduction, line 47: spelling mistake ‘thorough’. But in fact, what do the authors mean by trafficking ‘through’ the membranes?

- Introduction, lines 65-69. It is mentioned that ‘in the neo1Δcfs1Δ mutant, phospholipid asymmetry may not be normally regulated’. As cfs1 is a suppressor of neo1, it does not sound so obvious. Why should we envision defects in the neo1Δcfs1Δ mutant? Where does this assumption originate from? I think it would help the general reader to explain a bit better the rationale followed here. This is further substantiated by the fact that at lines 157-158 in the results section, it is claimed that the cfs1Δ mutation completely suppresses PE and PS exposure.

- Results, lines 182-184. The sentence is odd, as it starts by saying that Pdr5 accumulated in the Neo1-depleted cfs1Δerd1Δ mutant, and finishes with ‘although the accumulation of Pdr5-GFP was relatively low’. In addition, it’s not really clear form the figure that the accumulation is low.

- Results, lines 184-185. What is the evidence at this stage that the defect in protein trafficking is in the secretory rather than in the endocytic pathway? Especially given that Drs2/Cdc50 mutations interfere with endocytic retrieval of Snc1?

- This is probably a very naive question, which is however relevant to several experiments displayed in the manuscript: in the legend to figure 1B, it is mentioned that cells are grown in YPDA, and then plated either on YPDA or YPGA; however, as Neo1 is an essential gene, how can cells grow with Neo1 under the control of a GAL promoter in a medium containing glucose as a carbon source?

- Legend to figure 1C, line 198. It is mentioned that ‘strains are cultured for 20h…’ In which medium are cells cultured?

- Results, line 248. ‘…the TGN is exposed to the accumulated PS’. The wording sounds odd to me if the aim is to say that PS is exposed to the cytosolic leaflet of the TGN. This wording can be found elsewhere in the manuscript. Moreover, how does PS distribute, using Lact-C2 and evt-2 probes, in Neo1-depleted cfs1Δerd1Δ mutant?

- Results, line 264. It is suggested that absence of PI4P in the TGN may be the reason for membrane trafficking defects in the Neo1-depleted cfs1Δerd1Δ mutant. Isn’t it also true for the neo1Δ mutant? I suggest it’s worth recalling it so that a connection is made between the Neo1 flippase and PI4P.

- Title of Fig 3. It is claimed that the TGN ‘has no PI4P’. I guess it would be fairer to say that PI4P levels dropped dramatically in the TGN, in the absence of sensitive assays to measure PI4P levels.

- In my opinion, the manuscript would gain in clarity if a simple schematic recalling the direction of phosphate transport envisioned for Erd1 and demonstrated for PHO transporters were added. It’s not immediately obvious that PHO transporters are involved in the uptake of phosphate and that would help understand the rationale of the experiments (lines 333-336).

- Results, line 342. What is the phosphate concentration in a ‘normal’ SD medium (Fig 5B)?

- Results, lines 360-361. It’s probably a bit bold to claim that the authors demonstrated that ‘…Pho87 and Pho90 accelerated transport of the luminally accumulated Pi to the cytoplasm…’ in the absence of transport assays.

- Legend to Fig 6A, line 409. Why the authors do not provide statistic tests here as for the other figures (figure 4 for instance)? Is the observed difference significant?

- Results, lines 427-428. I’m not sure about the wording of the sentence. Is it the induction of UPR by DTT in the WT which is referred to? Why not comparing induction of UPR in the Neo1-depleted cfs1Δerd1Δ mutant with that of the WT in the absence of DTT? And the induction of UPR in the Neo1-depleted cfs1Δerd1Δ mutant does not seen higher than that of WT +DTT?

6. PLOS authors have the option to publish the peer review history of their article (what does this mean?). If published, this will include your full peer review and any attached files.

Reviewer #1: No

Reviewer #2: Yes: Guillaume Lenoir

---

## [Author Response · Author response to Decision Letter 0]

9 Jun 2020

We thank the reviewers for valuable comments on our manuscript. We have thoroughly revised our manuscript according to the reviewers’ comments as described below.

Our responses to the comments of the Reviewer #1

1) I have trouble grasping the meaning of some key statements in the manuscript. For instance, in the Abstract and on p. 4 (bottom) and p. 26 (middle), the authors state “our results suggest that flippase-mediated phospholipid redistribution is functionally vital not only in the cytoplasmic leaflet but also in the luminal leaflet of the Golgi membrane”. I do not see the added value of such a cryptic statement. The study also does not provide any concrete data on the lipid composition of the luminal leaflet in the Golgi of wildtype and mutant cells. I urge the authors to formulate a less ambiguous take-home message, one that focuses more on their principal experimental findings.

We agree to the reviewer’s comments that our data do not lead us to any conclusion concerning a functional difference between the cytoplasmic leaflet and the luminal leaflet of the Golgi membrane. According to the reviewer’s suggestion, we have changed the sentences to “Our results suggest that phospholipid asymmetry and luminal Pi concentration coordinately regulate Golgi functions” in the Abstract (lines 32-33), the Introduction (lines 71-72), and the Discussion (lines 494-495). 

2) On p. 19 (bottom) and p. 20 (top), the authors state that their findings demonstrate that overexpression of Pho87p and Pho90p accelerates transport of luminally accumulated Pi to the cytoplasm, and conclude that elevated Pi levels in the Golgi lumen are responsible for the lethality of the Neo1p-depleted csf1/erd1 mutant cells. However, their experimental data provide only indirect evidence for this. Therefore, they should significantly tune down these claims, for instance by replacing the words “demonstrate” for “suggest” and “conclude” for “propose”.

We agree to the reviewer’s comment that we should tone down our statement on the significance of elevated Pi levels in the Golgi lumen for the lethality of the Neo1p-depleted cfs1 erd1 mutant. According to the reviewer’s suggestion, we have replaced the words “demonstrate” for “suggest” and “conclude” for “propose” (lines 362-365). 

3) To dissect the complex genetic interactions that implicate phospholipid asymmetry and phosphate homeostasis in regulating Golgi function, the authors analyzed the consequences of Neo1p depletion on Golgi function in wildtype, csf1 and csf1/erd1 mutant cells. As Erd1p has been suggested to transport Pi from the lumen of the Golgi, analyzing also the consequences of Neo1p depletion in csf1 mutant cells on Golgi functions would have been particularly informative. However, such experiments are not part of the present study. Why not?

Although the reviewer suggests “analyzing the consequences of Neo1p depletion in csf1 mutant”, we already analyzed this mutant in the original version. Judging from the context, we think that the reviewer suggests “analyzing the consequences of Neo1p depletion in erd1 mutant”. However, because the Neo1p depletion in wild-type cells shows severe defects in Golgi functions, the Neo1p-depleted erd1Δ mutant would show similar severe defects unless the erd1Δ mutation suppresses the neo1Δ mutation. We actually constructed the Neo1p-depleted erd1Δ mutant expressing GFP-Snc1p-pm. This mutant exhibited massive accumulation of GFP-Snc1p-pm as the Neo1p-depleted and Neo1p-depleted cfs1Δ erd1Δ mutants. This result has been shown in Figure 1C in the revised version.

4) p. 14 (top) “…the TGN is exposed to the accumulated PS” should read “…PS is accumulated in the cytosolic leaflet of the TGN”

5) p. 14 (end 1st para) “…the cytoplasmic leaflet of the TGN…. is exposed to PS” should be reformulated as indicated above.

6) Fig. 3 caption “The TGN… is exposed to PS and has no PI4P” should read “The cytoplasmic leaflet of the TGN … contains PS and has no PI4P”

We thank the reviewer for pointing out our mistakes. We have corrected these points accordingly (lines 250, 254-255, 276, and 278).

Our responses to the comments of the Reviewer #2

- In the abstract (lines 32-34) and in the discussion (lines 488-489), the authors propose that phospholipid redistribution is not only important in the cytosolic leaflet but also in the luminal leaflet. Do the authors mean that phospholipid asymmetry has an important impact not only on the cytosolic face of the membrane but also on the luminal one? I find the word ‘leaflet’ misleading in this case, as it restricts the impact of phospholipid asymmetry to the membrane itself

This comment has also been made by the Reviewer #1. We agree to the reviewer’s comments that our data do not lead us to any conclusion concerning a functional difference between the cytoplasmic leaflet and the luminal leaflet of the Golgi membrane. According to the reviewer’s suggestion, we have changed the sentences to “Our results suggest that phospholipid asymmetry and luminal Pi concentration coordinately regulate Golgi functions” in the Abstract (lines 32-33), the Introduction (lines 71-72), and the Discussion (lines 494-495). 

- Introduction, line 47: spelling mistake ‘thorough’. But in fact, what do the authors mean by trafficking ‘through’ the membranes?

We thank the reviewer for pointing out our mistake. As questioned by the reviewer, we do not mean any specific membrane trafficking pathway by “through”. Therefore, this word has been deleted. (line 47)

- Introduction, lines 65-69. It is mentioned that ‘in the neo1Δcfs1Δ mutant, phospholipid asymmetry may not be normally regulated’. As cfs1 is a suppressor of neo1, it does not sound so obvious. Why should we envision defects in the neo1Δcfs1Δ mutant? Where does this assumption originate from? I think it would help the general reader to explain a bit better the rationale followed here. This is further substantiated by the fact that at lines 157-158 in the results section, it is claimed that the cfs1Δ mutation completely suppresses PE and PS exposure.

As pointed out by the reviewer, our statement that “in the neo1Δ cfs1Δ mutant, phospholipid asymmetry may not be normally regulated” may not sound obvious. To make the sentence more consistent to readers, we have changed this sentence to “However, the neo1Δ cfs1Δ mutant may not be equivalent to the wild type in its Golgi functions, otherwise a set of these two genes including the essential NEO1 gene would be dispensable.” (lines 65-67)

- Results, lines 182-184. The sentence is odd, as it starts by saying that Pdr5 accumulated in the Neo1-depleted cfs1Δerd1Δ mutant, and finishes with ‘although the accumulation of Pdr5-GFP was relatively low’. In addition, it’s not really clear form the figure that the accumulation is low.

We thank the reviewer for these comments. To make how the accumulation of Pdr5-GFP was relatively low clearer, we have modified the sentence as follows. “…, although the accumulation of Pdr5p-GFP was relatively low compared to GFP-Snc1p-pm and GFP-Sso1p” (line 184). As to Fig 1D, the number of cells that showed clear accumulation of Pdr5p-GFP was lower compared to GFP-Snc1p-pm and GFP-Sso1p, but the cells that showed accumulation accumulated Pdr5p-GFP to an extent similar to those of GFP-Snc1p-pm and GFP-Sso1p.

- Results, lines 184-185. What is the evidence at this stage that the defect in protein trafficking is in the secretory rather than in the endocytic pathway? Especially given that Drs2/Cdc50 mutations interfere with endocytic retrieval of Snc1?

Because we used GFP-Snc1p-pm, which is a mutant version that is not endocytosed, this protein does not enter the endocytic recycling pathway. However, it is known that Pdr5p is endocytosed and delivered to vacuoles. Thus, we have changed the sentence to “These results suggest that the Neo1p-depleted cfs1Δ erd1Δ mutant exhibits major defects in membrane trafficking pathways.” (lines 184-186)

- This is probably a very naive question, which is however relevant to several experiments displayed in the manuscript: in the legend to figure 1B, it is mentioned that cells are grown in YPDA, and then plated either on YPDA or YPGA; however, as Neo1 is an essential gene, how can cells grow with Neo1 under the control of a GAL promoter in a medium containing glucose as a carbon source?

Because the catabolite repression is slow in the GAL1 promoter, and because the GAL1 promoter is a strong promoter, we need preincubation of the cells in YPDA medium to deplete Neo1p for ~12 h. Otherwise, the PGAL1-NEO1 cells grown in YPGA continue to grow for ~12 h even in the YPDA medium. We have added “to deplete Neo1p” in the revised version (lines 193 and 199)

- Legend to figure 1C, line 198. It is mentioned that ‘strains are cultured for 20h…’ In which medium are cells cultured?

As written above, the cells were cultured in YPDA medium to deplete Neo1p. In the sentence pointed out by the reviewer, there was a redundant description that indicated that the cells were cultured more than 20 h in YPDA. This description, “and were cultured”, which is our mistake, has been deleted in the revised version (line 199). Because the repression of the GAL1 promoter is not 100% in YPDA medium, some mutant requires depletion time of more than 12 hours. As shown in Fig 1B, the PGAL1-NEO1 cfs1Δ erd1Δ mutant showed some residual growth compared to the PGAL1-NEO1 mutant, suggesting that the 12 h depletion is not enough to completely inhibit cell growth of the PGAL1-NEO1 cfs1Δ erd1Δ mutant. To observe the terminal phenotype of the PGAL1-NEO1 cfs1Δ erd1Δ mutant, we determined that twenty hours incubation in YPDA is appropriate for the PGAL1-NEO1 cfs1Δ erd1Δ mutant (line 199). 

- Results, line 248. ‘…the TGN is exposed to the accumulated PS’. The wording sounds odd to me if the aim is to say that PS is exposed to the cytosolic leaflet of the TGN. This wording can be found elsewhere in the manuscript. Moreover, how does PS distribute, using Lact-C2 and evt-2 probes, in Neo1-depleted cfs1Δerd1Δ mutant?

These our mistakes were also pointed out by the reviewer #1, and all of them have been corrected appropriately to describe that PS is exposed to the cytoplasmic leaflet of the TGN (lines 250, 254-255, 276, and 278). As to the PS distribution using Lact-C2 and evt-2 probes, these probes were similarly colocalized with GFP-Snc1p-pm in the Neo1-depleted cfs1Δ erd1Δ mutant (Fig 3A and B). As shown in Fig 2C, GFP-Snc1p-pm was colocalized with Sec7p-mRFP1. These results suggest that PS is exposed to the cytoplasmic leaflet of the TGN in the Neo1-depleted cfs1Δ erd1Δ mutant (lines 254-255). 

- Results, line 264. It is suggested that absence of PI4P in the TGN may be the reason for membrane trafficking defects in the Neo1-depleted cfs1Δerd1Δ mutant. Isn’t it also true for the neo1Δ mutant? I suggest it’s worth recalling it so that a connection is made between the Neo1 flippase and PI4P.

We thank the reviewer for pointing out that the Neo1p-depleted mutant should be included in the sentence. It is important to describe that the absence of PI4P may cause membrane trafficking defects in the Neo1p-depleted mutant. It has been added to the revised version (line 266).

- Title of Fig 3. It is claimed that the TGN ‘has no PI4P’. I guess it would be fairer to say that PI4P levels dropped dramatically in the TGN, in the absence of sensitive assays to measure PI4P levels.

We agree to the reviewer’s comment that we should not describe “has no PI4P”. According to the reviewer’s suggestion, we have changed it to “decreased level of PI4P” (line 277).

- In my opinion, the manuscript would gain in clarity if a simple schematic recalling the direction of phosphate transport envisioned for Erd1 and demonstrated for PHO transporters were added. It’s not immediately obvious that PHO transporters are involved in the uptake of phosphate and that would help understand the rationale of the experiments (lines 333-336).

We thank the reviewer for the comment. According to the reviewer’s suggestion, we have added the schematic model as Fig 5A. In addition, we have also added Fig 5F to explain the other model, which is consistent with our data.

- Results, line 342. What is the phosphate concentration in a ‘normal’ SD medium (Fig 5B)?

Phosphate concentration in SD medium is 7.3 mM according to the manufacturer. This has been added to Fig 5C and described in the legend in the revised version (lines 377 to 378).

- Results, lines 360-361. It’s probably a bit bold to claim that the authors demonstrated that ‘…Pho87 and Pho90 accelerated transport of the luminally accumulated Pi to the cytoplasm…’ in the absence of transport assays.

This comment has also been made by the reviewer #1. According to the reviewers’ suggestion, we have replaced the word “demonstrate” for “suggest” (line 362).

- Legend to Fig 6A, line 409. Why the authors do not provide statistic tests here as for the other figures (figure 4 for instance)? Is the observed difference significant?

According to the reviewer’s suggestion, we have performed statistic tests for Fig 6A (line 418) and Fig 1C (lines 204-205). Both tests gave a significant difference to the results as shown in the revised version.

- Results, lines 427-428. I’m not sure about the wording of the sentence. Is it the induction of UPR by DTT in the WT which is referred to? Why not comparing induction of UPR in the Neo1-depleted cfs1Δerd1Δ mutant with that of the WT in the absence of DTT? And the induction of UPR in the Neo1-depleted cfs1Δerd1Δ mutant does not seen higher than that of WT +DTT?

As pointed out by the reviewer, to compare the induction of UPR in the Neo1-depleted cfs1Δ erd1Δ mutant with that of the WT in the absence of DTT, we have deleted the words “compared to that induced by DTT” in the revised version (line 436).

---

## [Decision Letter · Decision Letter 1]

30 Jun 2020

PONE-D-20-06448R1

A complex genetic interaction implicates that phospholipid asymmetry and phosphate homeostasis regulate Golgi functions

PLOS ONE

Dear Dr. Tanaka,

Thank you for submitting your manuscript to PLOS ONE. After careful consideration, we feel that it has merit but does not fully meet PLOS ONE’s publication criteria as it currently stands. Therefore, we invite you to submit a revised version of the manuscript that addresses the points raised during the review process.

Two reviewers' evaluations are now in as shown below. I am happy to inform you that two reviewers #1 and #2 suggest minor revisions. Please read carefully those arguments and revise the manuscript item-by-item.

We look forward to receiving your revised manuscript.

Kind regards,

Reiko Sugiura, M.D., PhD.

Academic Editor

PLOS ONE

Reviewers' comments:

Reviewer's Responses to Questions

**Comments to the Author**

1. If the authors have adequately addressed your comments raised in a previous round of review and you feel that this manuscript is now acceptable for publication, you may indicate that here to bypass the “Comments to the Author” section, enter your conflict of interest statement in the “Confidential to Editor” section, and submit your "Accept" recommendation.

Reviewer #1: All comments have been addressed

Reviewer #2: All comments have been addressed

2. Is the manuscript technically sound, and do the data support the conclusions?

Reviewer #1: Yes

Reviewer #2: Yes

3. Has the statistical analysis been performed appropriately and rigorously? 

Reviewer #1: Yes

Reviewer #2: Yes

4. Have the authors made all data underlying the findings in their manuscript fully available?

Reviewer #1: Yes

Reviewer #2: Yes

5. Is the manuscript presented in an intelligible fashion and written in standard English?

Reviewer #1: Yes

Reviewer #2: Yes

6. Review Comments to the Author

Reviewer #1: I urge the authors to have the manuscript proof-read by a native English speaker.

xxxxxxxxxxxxxxxx

Reviewer #2: The authors adequately addressed the issues I have raised.

- However, I think it would help indicating the (perhaps putative) function of the genes which mutations are involved in ER retention (line 174), for instance in the legend to Figure S1A. The same holds for Fig S1B.

- Line 205: '...fluorescent microscope' should read '...fluorescence microscope'?

- Please indicate in the legend to Fig 5A what the rod-shaped grey drawing corresponds to.

7. PLOS authors have the option to publish the peer review history of their article (what does this mean?). If published, this will include your full peer review and any attached files.

Reviewer #1: No

Reviewer #2: **Yes: **Guillaume Lenoir

---

## [Author Response · Author response to Decision Letter 1]

4 Jul 2020

Our responses to the comments of the Reviewer #1

I urge the authors to have the manuscript proof-read by a native English speaker.

The unrevised original version of our manuscript had already been checked by a leading company for English language editing. However, their work does not seem to be perfect, because the sentences pointed out by the reviewer #1 (e.g. comments 4, 5, and 6), which were also pointed out by the reviewer #2, were modified according to the editor’s suggestions. In the revised version, we finally made these sentences back almost to the original ones. We understand that this is because the editor would not be a specialist in our filed. Therefore, in the revised version, we checked the manuscript by ourselves for any kind of mistakes as much as possible. 

Our responses to the comments of the Reviewer #2

- However, I think it would help indicating the (perhaps putative) function of the genes which mutations are involved in ER retention (line 174), for instance in the legend to Figure S1A. The same holds for Fig S1B.

According to the reviewer’s suggestion, gene (protein) functions have been included in S1 Fig A and B.

- Line 205: '...fluorescent microscope' should read '...fluorescence microscope'?

We thank the reviewer for pointing out our mistake. It has been corrected in the revised version (line 200).

- Please indicate in the legend to Fig 5A what the rod-shaped grey drawing corresponds to.

According the reviewer’s suggestion, we have indicated in the legend that the rod-shaped grey drawing corresponds to membrane proteins to be transported to the plasma membrane (line 371).

---

## [Editor Report · Decision Letter 2]

9 Jul 2020

A complex genetic interaction implicates that phospholipid asymmetry and phosphate homeostasis regulate Golgi functions

PONE-D-20-06448R2

Dear Dr. Tanaka,

We’re pleased to inform you that your manuscript has been judged scientifically suitable for publication and will be formally accepted for publication once it meets all outstanding technical requirements.

Kind regards,

Reiko Sugiura, M.D., PhD.

Academic Editor

PLOS ONE

---

## [Editor Report · Acceptance letter]

15 Jul 2020

PONE-D-20-06448R2 

A complex genetic interaction implicates that phospholipid asymmetry and phosphate homeostasis regulate Golgi functions 

Dear Dr. Tanaka:

I'm pleased to inform you that your manuscript has been deemed suitable for publication in PLOS ONE. Congratulations! Your manuscript is now with our production department. 

Kind regards, 

on behalf of

Dr. Reiko Sugiura 

Academic Editor

PLOS ONE